# Reconstruction of the ancient cyanobacterial proto-circadian clock system KaiABC

Silin Li[1,10], Zengxuan Zhou[1,10], Yufeng Wan [ID][1,9,10], Xudong Jia [ID][1,10], Peiliang Wang [ID][1], Yu Wang[2], Taisen Zuo [ID][3,4], He Cheng[3,4], Xiaoting Fang[1], Shuqi Dong[5], Jun He[5], Yilin Yang[1], Yichen Xu[1], Shaoxuan Fu[1], Xujing Wang[1], Ximing Qin[6], Qiguang Xie [ID][2], Xiaodong Xu [ID][2], Yuwei Zhao[7], Dan Liang[8], Peng Zhang[8], Qinfen Zhang [ID][1] & Jinhu Guo [ID][1✉]

## Abstract

**Earlier in its history, the Earth used to spin faster than it does today. How ancient organisms adapted to the short day/night cycles during that time remains unclear. In this study we reconstruct and analyse the ancient circadian clock system KaiABC (anKaiABC) of cyanobacteria that existed ~0.95 billion years ago, when the daily light/dark cycle was ~18 h-long. Compared to their contemporary counterparts, anKaiABC proteins had different structures and interactions. The kinase, phosphatase, and adenosine triphosphatase (ATPase) activities of anKaiC were lower, while the anKaiA and anKaiB proteins were less effective at regulating the KaiC/anKaiC phosphorylation status. We provide evidence indicating that the anKaiABC system does not endogenously oscillate, but it can be entrained by an 18 h-long light/dark cycle. A *Synechococcus* strain expressing *ankaiABC* genes exhibits better adaptation to 9-h light/9-h dark cycles (LD9:9) that mimic the ancient 18-h day/night cycles, whereas the *kaiABC*-expressing strain preferentially adapts to the LD12:12 contemporary conditions. These findings suggest that, despite its lack of self-sustaining circadian oscillation, the proto-circadian system may have mediated adaptation of ancient cyanobacteria to the 18 h-long light/dark cycles present 0.95 billion years ago.**

**Keywords** Circadian Clock; Cyanobacteria; KaiABC; Adaptation; Evolution
**Subject Categories** Evolution & Ecology; Post-translational Modifications & Proteolysis

## Introduction

The Earth rotates on its axis, and as a consequence, many environmental factors, such as light, temperature, humidity, and radiation, periodically fluctuate. Numerous eukaryotic organisms possess sustainable circadian clock systems adapted to periods of circa 24 h, which endows them with the ability to synchronise their physiology and behaviour to light/dark cycle established by the Earth's rotation (Yerushalmi and Green, 2009). Circadian rhythms are also present in some prokaryotic species, such as cyanobacteria and nonphotosynthetic *Bacillus subtilis* (Eelderink-Chen et al, 2021; Johnson et al, 2017). It has been demonstrated in cyanobacteria and some other species, that a circadian clock with period close to the environmental light-dark cycling enhances fitness or facilitates adaptation (Ouyang et al, 1998; Woelfle et al, 2004).

Cyanobacteria are considered as the earliest organisms on Earth, potentially emerging up to 3.5 Ga ago (Schopf and Packer, 1987). Circadian rhythms in multiple physiological processes, for instance, nitrogen fixation and photosynthesis, have been found in several cyanobacteria species including *Synechococcus* RF-1, Miami BG 43511/43522, WH7803 and *Synechococcus elongatus* PCC7942. Specifically, *S. elongatus* PCC7942 has an endogenous circadian system that consists of KaiABC proteins. KaiC is both an autokinase and an autophosphatase that contains two phosphorylation sites: S431 and T432 (Nishiwaki et al, 2004). KaiC phosphorylation is correlated with the dual adenosine triphosphatase domains located in the CI (1–268 aa) and CII (269–519 aa) regions, respectively (Hayashi et al, 2004). KaiA binds to the KaiC hexamer and promotes KaiC phosphorylation, whereas KaiB binds to KaiC which sequesters KaiA from the KaiC A-loop (488–497 aa) (Iwasaki et al, 2002; Kim et al, 2008; Kitayama et al, 2003; Swan et al, 2022). Interestingly, even in vitro, the KaiABC proteins display intrinsic oscillation of KaiC phosphorylation and dephosphorylation (Nakajima et al, 2005).

[1]School of Life Sciences, Key Laboratory of Gene Engineering of the Ministry of Education, Sun Yat-sen University, Guangzhou, China. [2]State Key Laboratory of Crop Stress Adaptation and Improvement, School of Life Sciences, Henan University, Kaifeng, China. [3]Institute of High Energy Physics, Chinese Academy of Sciences, Beijing, China. [4]Spallation Neutron Source Science Center, Dongguan, China. [5]Center for Biomedical Digital Science, State Key Laboratory of Respiratory Disease, Guangdong Provincial Key Laboratory of Stem Cell and Regenerative Medicine, GIBH-CUHK Joint Research Laboratory on Stem Cell and Regenerative Medicine, Guangzhou Institutes of Biomedicine and Health, Chinese Academy of Sciences, Guangzhou, China. [6]Department of Health Sciences, Institutes of Physical Science and Information Technology, Anhui University, Hefei, China. [7]School of Life Sciences, Northwest University, Xi'an, China. [8]State Key Laboratory of Biocontrol, Guangdong Provincial Key Laboratory for Aquatic Economic Animals, School of Life Sciences, Sun Yat-Sen University, Guangzhou, China. [9]Present address: Department of Biology, Texas A&M University, College Station, TX, USA. [10]These authors contributed equally: Silin Li, Zengxuan Zhou, Yufeng Wan, Xudong Jia. ✉E-mail: guojinhu@mail.sysu.edu.cn

The evolutionary history of some cyanobacterial circadian genes, e.g., *kaiC*, *sasA*, *cikA* and *cpmA*, can be dated back to approximately 3.5 Ga ago. It has been proposed that KaiA might have appeared in cyanobacteria genomes 1.4–1.6 Ga ago or even earlier (Baca et al, 2010; Dvornyk, 2006). Some modern cyanobacteria, including *Gloeobacter violaceus* PCC 7421, *Prochlorococcus sp*. strain MED4, *Rhodopseudomonas palustris*, and *Rhodobacter spheroides*, contain no *kaiA*, suggesting that they have no endogenous self-sustaining circadian clocks (Axmann et al, 2009; Holtzendorff et al, 2008; Ma et al, 2016; Min et al, 2005).

The rotation of Earth has slowed over time, mainly because of gravitational interaction with the moon; Earth's rotation was much faster in ancient times, as evidenced by astronomical and palaeontological data. The ancient environment on Earth dramatically differed from the present environment, including the daily light/dark period, which was much shorter because of the faster rotation of the Earth'. For example, 0.9 Ga ago, the day length was ~18 h (Dvornyk, 2009; Scrutton and Hipkin, 1973; Sonett et al, 1996; Spalding and Fischer, 2019). An analysis of fossilized mollusc shells revealed that the ancient animals experienced days that were shorter than those today (Zhao et al, 2007). However, to date, there is very limited knowledge regarding this phenomenon at the molecular level.

Investigating the characteristics of ancient circadian systems and their contribution to adaptation to the ancient environment is very intriguing but challenging. In this study, we predicted and reconstructed an ancient cyanobacterial proto-circadian clock system that existed ~0.95 Ga ago, and the functional characteristics and adaptations to the daily light-dark cycles of the temporal environment were analysed.

# Results

## Phylogenetic analysis of cyanobacterial KaiABC proteins

Ancient proteins can be reconstructed and their biochemical properties and functions can be assayed on the basis of phylogenetic analysis of their evolutionarily contemporary descendants (Hochberg and Thornton, 2017). For example, Gaucher et al, reconstituted the ancient sequences for elongation factors of the Tu family (EF-Tu), and the ancient EF-Tu proteins show temperature preference of 55–65 °C, suggesting that the ancient bacteria harbouring these genes are thermophiles (Gaucher et al, 2003).

During the early history of the Earth, the planet rotated much faster, which means that the period of daily light/dark cycling was much shorter (Zhao et al, 2007). To investigate whether the ancient circadian clock was aligned with the ancient light/dark cycling environment, we obtained the KaiABC sequences of available extant cyanobacterial species and performed phylogenetical analyses with amino acid sequences of 9 KaiA proteins, 19 KaiB proteins, and 19 KaiC proteins from various cyanobacterial lineages covering *Synechococcus*, *Thermosynechococcus*, *Prochlorococcus marinus*, *Prochlorothrix*, *Arthrospira*, *Acaryochloris*, and *Cyanothece*. The candidate ancestral sequences were then reconstructed at nodes throughout the bacterial subtrees, and the results indicated that the ancestral KaiABC sequences have ancestors at the same time ~0.95 Ga ago on the basis of the available KaiABC sequences from modern cyanobacterial species (Fig. 1A–C). The divergence time was deduced according to the 16S rRNA gene information (Dvornyk et al, 2003). The overall sequences of critical motifs in anKaiABC proteins were highly conserved with those in KaiABC,

especially in KaiC, despite the dispersed amino acid residues (Fig. 1D–F). On the basis of these phylogenetic results and sequence alignments, we synthesised ancestral genes expressing these deduced ancestral KaiABC proteins (anKaiABC).

## Structural analysis of ancient clock proteins through cryogenic electron microscopy (cryo-EM)

anKaiABC proteins were expressed in prokaryotes and subsequently purified (Appendix Fig. S1A–H). We analysed the structure via cryo-EM single particle reconstruction, and obtained the structure of anKaiC with a resolution of 2.5 Å (Fig. EV1A–E; Table EV1). anKaiC can form a conserved homohexamer, which bears a highly conserved structure with KaiC (Fig. 2A) (PDB:7S67) (Swan et al, 2022).

Dispersed differences in the surface hydrophobicity and structure between KaiC and anKaiC were observed (Fig. 2B). The A-loop is the binding site for KaiA to facilitate KaiC phosphorylation (Egli et al, 2013). The sequences of the detectable part of the anKaiC A-loop (R492–T499, S-shaped loop) are almost identical to those in KaiC, and they have a very similar structure in this region (Fig. EV1F) (Swan et al, 2022). However, whether the structure of the remaining C-terminal part (R493–S524) of the A-loop is similar remains unclear because this region is so flexible that the structure cannot be resolved.

In KaiC the binding site for KaiB is called the B-loop (116–123 aa), and the KaiB-KaiC interaction also leads to disassociation of KaiA from the KaiC A-loop and binding with KaiB (Tseng et al, 2014). In this way, KaiA is sequestered from KaiC which causes its dephosphorylation (Egli et al, 2013). Differences in the B-loop (A112–D126) and another spatial loop (S150–A160) close to the B-loop were observed, although its function remains elusive (Fig. 2C).

anKaiC harbours two clefts for the binding of ATP molecules in its CI (KaiC-CI) and CII domains, similar to KaiC (Fig. 2D) (Nishiwaki et al, 2004). The KaiC-CI region exhibits extremely weak but stable ATPase activity (Terauchi et al, 2007). At the structural level, the extremely slow ATPase activity of the Kai-CI region is due to the sequestration of lytic water molecules from the γ-phosphate of ATP and the cis-to-trans isomerization of the peptide bond between residues D145 and S146 (Abe et al, 2015). We compared the structure of ATP-binding sites in the CI region between anKaiC and a recently published structure of full-length KaiC derived from cryo-EM analysis (PDB:7S67) (Swan et al 2022). Although the inside surface hydrophobicity of anKaiC and KaiC are almost identical (Fig. EV1G), the positions and orientations of the side chain amino acid residues (R165, E187, D206, and R230) surrounding the ATP molecule, showed subtle differences compared with the features of modern KaiC. Moreover, subtle differences in the position and orientation of ATP molecules were also observed (Fig. 2E–G). These structural discriminations may lead to functional differences in the catalytic ability of anKaiC as an ATPase, kinase or phosphatase.

## Small angle neutron scattering (SANS) analysis of KaiAC/anKaiAC proteins

SANS is a powerful tool for calibrating the dynamic conformation and interaction of macromolecules in solution (Zuo et al, 2024). The samples of KaiC, anKaiC, KaiA, and anKaiA, along with their mixtures, were analyzed via the SANS method. The neutron scattering curves for both KaiA and anKaiA were fitted via the polymer_excluded_volume model with the SasView software,

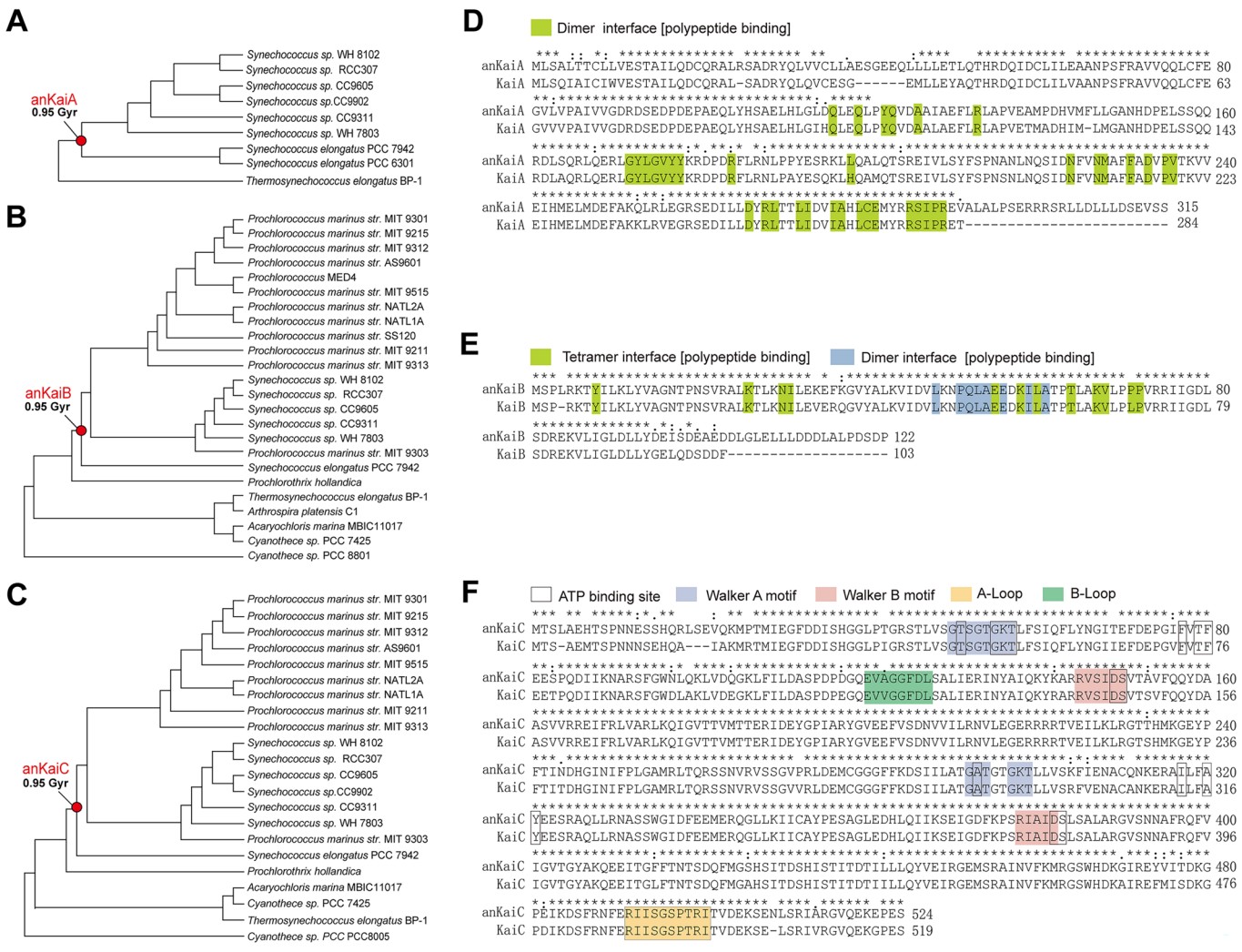

**Figure 1. Prediction of ancient KaiABC (anKaiABC) proteins.**

(A–C) Unrooted universal phylogenetic trees used to reconstruct ancestral KaiABC proteins. The branching time of ~0.95 Ga is denoted. (D–F) Protein sequence alignments between KaiA and anKaiA (D), KaiB and anKaiB (E), and KaiC and anKaiC (F) are shown. The critical domains are highlighted in different colours. Source data are available online for this figure.

which indicated a Porod factor close to 3, suggesting that neither protein is densely compact. The radius of gyration for KaiA was determined to be 3.4 nm, whereas it was 8.6 nm for anKaiA. Given their similar molecular weights, these findings suggest that multiple anKaiA molecules assemble together in the solvent. Additionally, the scattering curve for KaiA suggests a propensity for assembly, or partial aggregation into temporary clusters, as indicated by the low-Q upturn in Fig. 2H. KaiB and anKaiB proteins were precluded for SANS analysis due to precipitation in sample preparation.

The scattering curves of KaiC and anKaiC were both fitted via the dumbbell model featuring a central hole along the axis (Fig. EV2A–C). The significant upturn at very low Q in KaiC suggests a stronger tendency for KaiC to cluster (Fig. 2I). Half of the molecules of KaiC may have already clustered since the scale factor or the density of nonclustered scatterers within anKaiC sample is twice that of KaiC. Fitting results showed that the size of anKaiC is about 9% longer than that of KaiC. Both of the mixed samples KaiAC and anKaiAC showed low Q upturn signaturing the

clustering tendency. However, the dumbbell structure of anKaiAC is almost preserved according to the fitting results while the slope of KaiAC curve changed from −4 to −3 around 0.05 Å⁻¹ and the peak around 0.1 Å⁻¹ was smoother, implying that KaiA may be absorbed on the surface of KaiC (Fig. 2J). These data suggest a more prominent interaction between KaiA and KaiC compared to that of anKaiA and anKaiC, and the activity of anKaiA in promoting KaiC phosphorylation may be lower than that of KaiA.

## Functional characterization of ancient KaiABC proteins

Cyanobacterial KaiC is both an autokinase and a phosphatase. At high temperatures, KaiC undergoes dephosphorylation (Pattanayek et al, 2009). Similarly, anKaiC also showed a decrease in phosphorylation at 30 °C but the rate was much lower (Fig. 3A). Shifting to a low temperature (4 °C) after exposure to 30 °C for 15 h led to a significant increase in the levels of both phosphorylated KaiC and anKaiC, although the increase in phosphorylated anKaiC levels was significantly lower

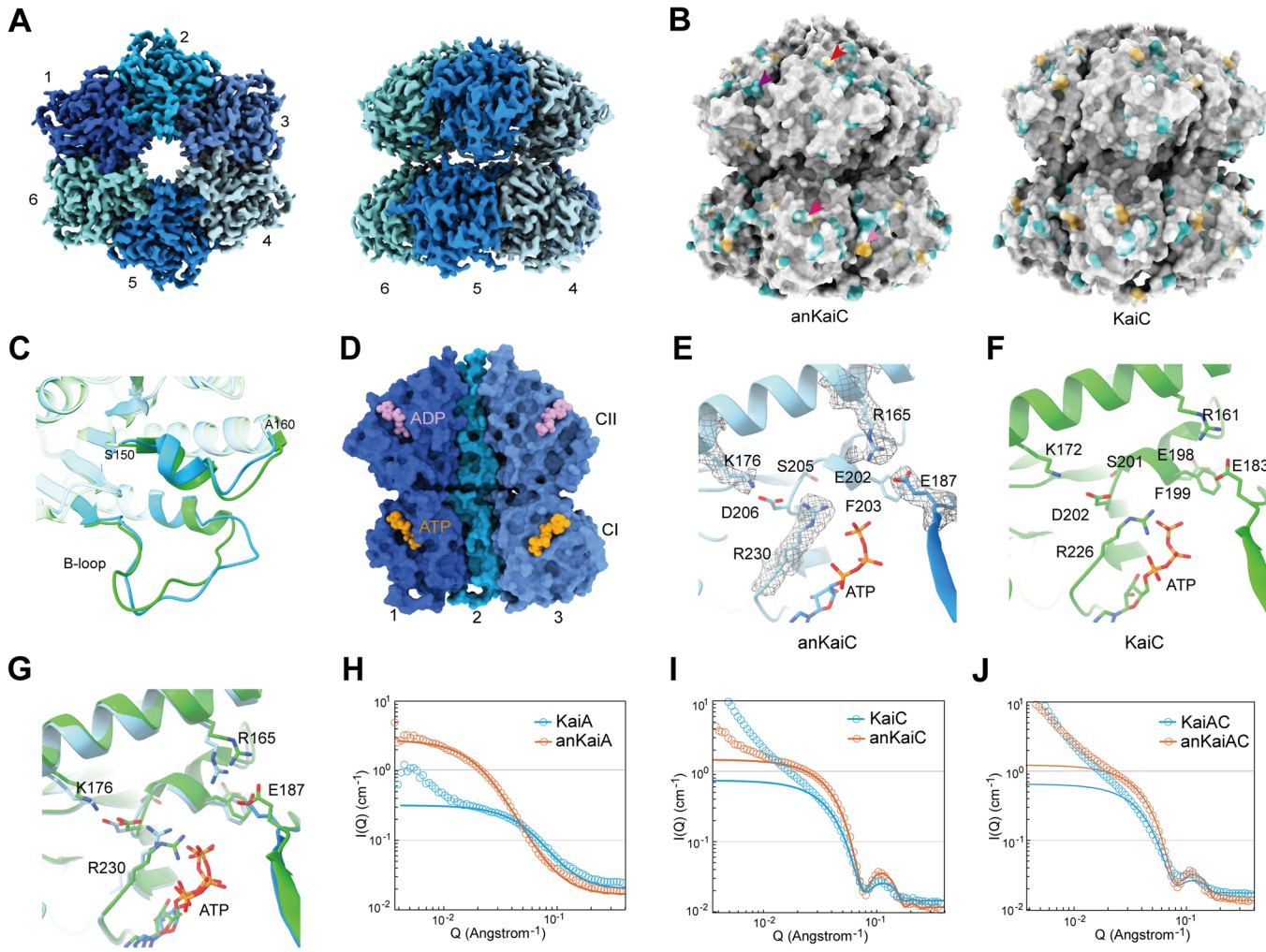

**Figure 2. Characterization of anKaiABC proteins.**

(A) Density map of the anKaiC hexamer viewed from the top (left panel) and side (right panel). The six anKaiC monomers are marked by numbers and coloured with different colours. (B) Structural differences between anKaiC (8JON) and KaiC (PDB: 7S67). The surface (outside view) with identical amino acid residues is shown in grey, the different amino acids between anKaiC and KaiC with hydrophilic residues are shown in cyan, and the hydrophobic residues are shown in yellow. The arrows in different colours denote the corresponding areas with different hydrophobicity due to different amino acid residues. (C) Alignment of models of anKaiC and KaiC (PDB:7S67) indicates that both the B-loop and the loop (S150–A160) of anKaiC do not align well with those of KaiC. (D) Interface representation of anKaiC (including monomers 1, 2 and 3) viewed from inside, with ATP nucleotides displayed in orange and ADP in pink. (E, F) The amino acids of anKaiC (E) and KaiC (F) surrounding the ATP molecule. (G) Alignment of models of anKaiC and KaiC (PDB:7S67) indicating that several side chains of the amino acids around the ATP molecule have subtle differences in location and orientation. (H–J) SANS analysis of KaiA and anKaiA (H), KaiC and anKaiC (I), and their mixtures kaiAC and anKaiAC (J). KaiA/anKaiA are fitted by the poly_excl_vol model in SASView and KaiC/anKaiC are fitted by the dumbbell model. Circles denote the SANS data and lines denote the data fit. Source data are available online for this figure.

than of the increase in phosphorylated KaiC levels (Fig. 3A). These data suggest that anKaiC has lower autokinase and phosphatase activities.

KaiA and anKaiA were mixed with KaiC/anKaiC, respectively, to assess the ability of anKaiA in promoting KaiC/anKaiC phosphorylation, and the results revealed that anKaiA failed to promote anKaiC phosphorylation in contrast to the significant effects of KaiA on KaiC (Fig. 3B). Next, we assessed the role of anKaiB in anKaiC dephosphorylation, and we found that, compared with KaiB, anKaiB was unable to inhibit anKaiC dephosphorylation. With the combination of KaiC and KaiA (Fig. 3C), the addition of anKaiB led to decreased KaiC phosphorylation, but its ability to facilitate KaiC dephosphorylation was lower than that of KaiB (Fig. 3D). We next compared the ATPase activities (ATP hydrolysis) of anKaiC and KaiC. The production of ADP was

monitored by high performance liquid chromatography (HPLC) method (Terauchi et al, 2007), and the results showed that the ATPase activity of anKaiC was significantly lower than that of KaiC (Fig. 3E). These data suggest that ancient cyanobacterial clock components are less efficient at modulating KaiC phosphorylation, dephosphorylation, and ATP hydrolysis than their contemporary counterparts are.

## Functional analysis of critical amino acid residue mutations on anKaiC/KaiC proteins

In B-loop and the spatially adjacent regions, the residue anKaiC-A122 is different from the same relative residue V118 in KaiC (PDB:5N8Y, Snijder et al, 2017) (Fig. 1F). We generated the V118A

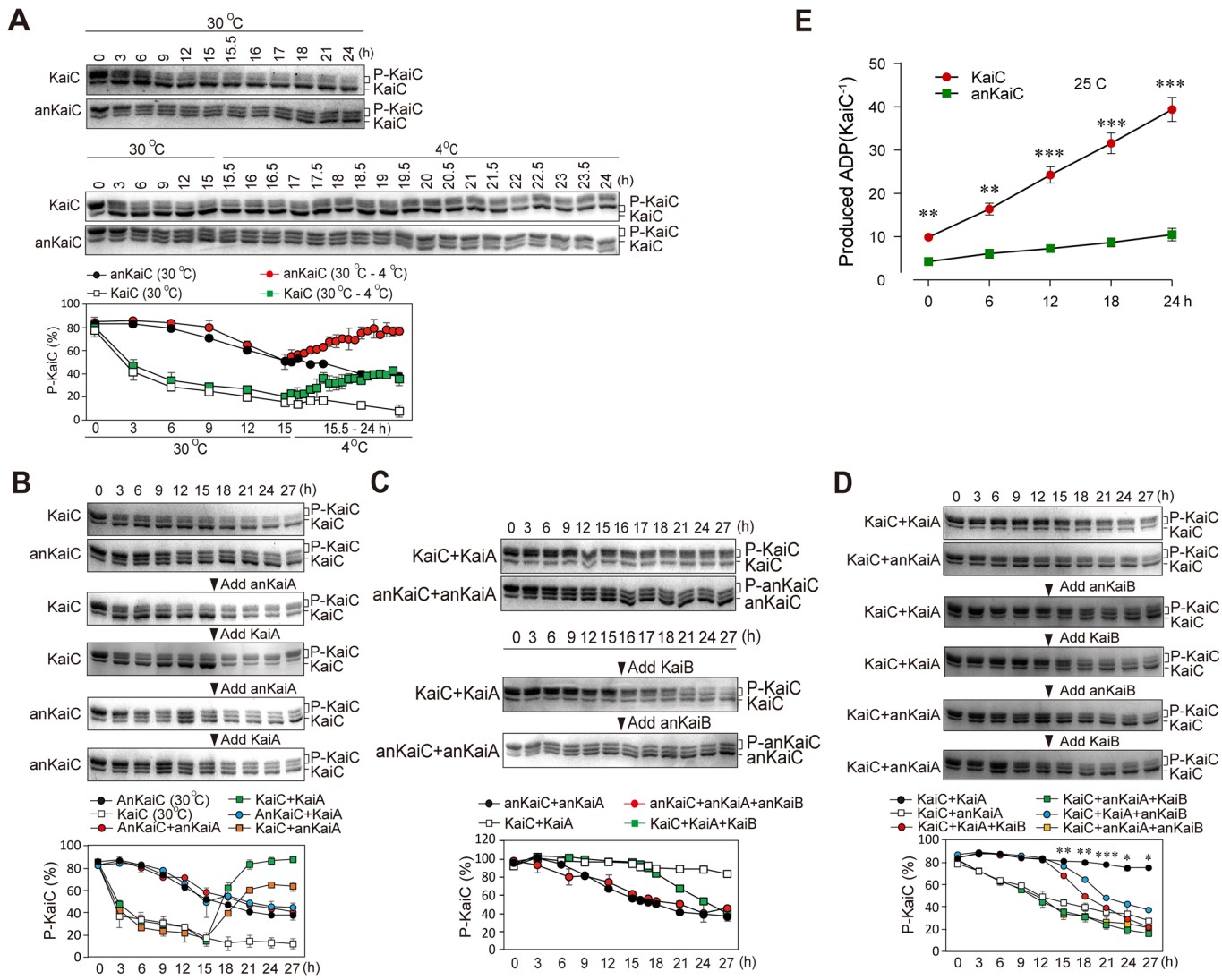

**Figure 3. Biochemical characterization of ancient KaiABC proteins.**

(A) Comparison of the autokinase activities between anKaiC and KaiC. (B) Function of anKaiA in promoting anKaiC/KaiC phosphorylation. (C) Function of anKaiB in promoting anKaiC/KaiC dephosphorylation. (D) Reciprocal assays of anKaiB/KaiB in promoting anKaiC/KaiC dephosphorylation. The results are visualized with Coomassie brilliant blue staining. Asterisks denote significant differences between the combinations of KaiA+KaiB+KaiC and KaiA+anKaiB+KaiC. *P* values from left to right: **P = 0.0065, **P = 0.0015, ***P = 0.0005, *P = 0.0409, *P = 0.0477. (E) ATPase assay results of KaiC and anKaiC. The ADP concentration in the reaction mixture was measured via LC-MS as described previously. *P* values from left to right: **P = 0.0014, **P = 0.0010, ***P = 0.0002, ***P = 0.0001, ***P <0.0001. Data are means ± SE (*n* = 3, independent experiments). Two-tailed unpaired Student's *t* test. Source data are available online for this figure.

mutant in KaiC (KaiC-V118A) (Fig. 4A; Appendix Fig. S2A,B), and KaiC protein bearing KaiC-V118A mutation was incubated with KaiA then KaiB was added at 30 °C (Fig. 4B), or transited from 4 °C to 30 °C (Fig. 4C), to assess the function in regulating the status of KaiC phosphorylation. In both experiments, the KaiC-V118A mutation showed no significant effects on the dephosphorylation process. Unexpectedly, this mutation led to compromised KaiC phosphorylation with the addition of KaiA (Fig. 4B,C).

In the CI ATP-binding and the spatially adjacent regions of anKaiC, the residues T233 and N244 are different from those on KaiC (S229 and T240 on KaiC) (PDB:5N8Y, Snijder et al, 2017) (Figs. 2G and 4A), suggesting these residues may contribute to the different activity in regulating protein interaction, KaiC phosphorylation and ATP hydrolysis. KaiC mutations

including S229T, T240N, and combination of S229T/T240N, were expressed and purified (Appendix Fig. S2C–G). The double mutations S229T/T240N compromised the dephosphorylation process although the individual mutations S229T, T240N showed no significant change compared to KaiC at constant 30 °C (Fig. 4D). After transition from 30 °C to 4 °C, the double mutations S229T/T240N showed a lower dephosphorylation level while the respective mutations of S229T or T240N showed comparable phosphorylation levels to that of KaiC (Fig. 4E).

The CI ATPase activity correlates with KaiC phosphorylation and circadian period length (Abe et al, 2015). The mutation S229T caused slight but significant alteration in the catalytic activity, while the mutation T240N dramatically enhanced the catalytic activity. In contrast, the activity of ATP hydrolysis was significantly repressed

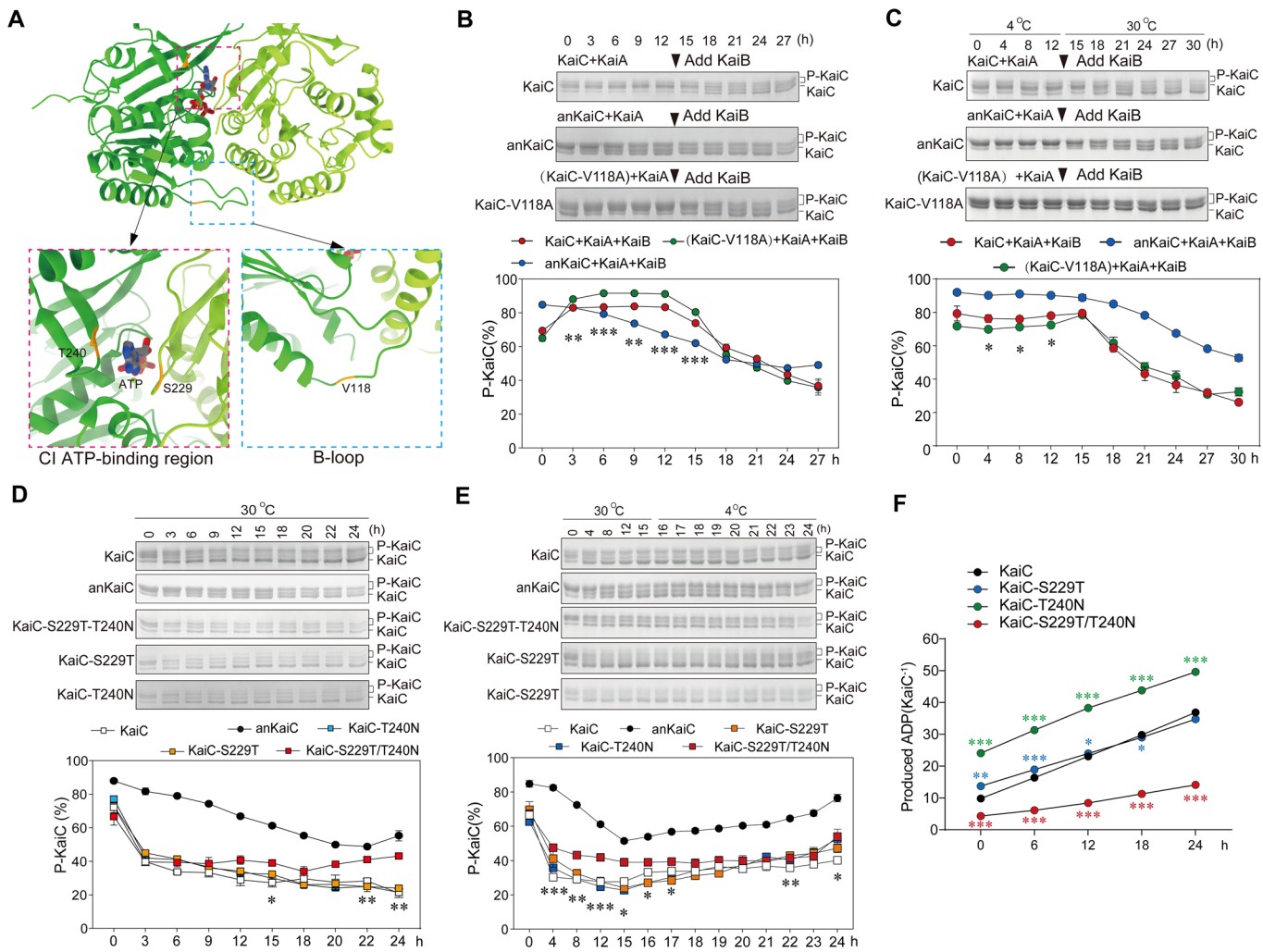

**Figure 4. Identification of critical residues contributing to KaiC/anKaiC difference.**

(A) Top panel: side view of two adjacent monomers of KaiC (PDB:7S67) CI domain, colored in green and light green, respectively. The ATP molecule is showed in stick style. Residues tested in point mutation experiments are labeled in orange. Bottom panel: enlarged views of the regions containing mutated amino acid residues. The magenta boxed area indicates the residues T240 and S229 which are spatial proximity to ATP molecule (left), and the blue boxed area indicates the residue V118 located in the B-loop (right). (B) Effects of mutations of KaiC-V118A on KaiC dephosphorylation constantly at 30 °C. Asterisks denote significances between KaiC and KaiC-V118A, P values from left to right: **P = 0.0018, ***P = 0.0004, **P = 0.0017, ***P = 0.0008, ***P = 0.0007. (C) Effects of mutations of KaiC-V118A on KaiC dephosphorylation with temperature transition. Asterisks denote significances between KaiC and KaiC-V118A, P values from left to right: *P = 0.0485, *P = 0.0392, *P = 0.0158. (D) Effects of mutations in ATP-binding region on maintaining KaiC phosphorylation at constant 30 °C. Asterisks denote significances between KaiC and KaiC-S229T/T240N, P values from left to right: *P = 0.0203, **P = 0.0064, **P = 0.0023. (E) Effects of mutations in ATP-binding region on maintaining KaiC phosphorylation at 30 °C and subsequent transition to 4 °C. Asterisks denote significances between KaiC and KaiC-S229T/T240N, P values from left to right: ***P = 0.0010, **P = 0.0016, ***P = 0.0002, *P = 0.0153, *P = 0.0322, *P = 0.0457, **P = 0.0045, *P = 0.0354. (F) ATPase assay results of KaiC proteins bearing mutations in the ATP-binding region. Asterisks denote significances between KaiC and mutant proteins in the same colours to the markers, P values in green from left to right: ***P <0.0001, ***P <0.0001, ***P <0.0001, ***P <0.0001, ***P <0.0001; P values in blue from left to right: **P = 0.0018, ***P = 0.0004, *P = 0.0460, *P = 0.0481; P values in red from left to right: ***P = 0.0005, ***P <0.0001, ***P <0.0001, ***P <0.0001, ***P <0.0001. Data are means ± SE (n = 3, independent experiments). Two-tailed unpaired Student's t test. Source data are available online for this figure.

in the double mutants (Fig. 4F). Together, these results demonstrate that these critical residues contribute to the different structural and functional properties between anKaiC and KaiC.

## Characterization of the ancient cyanobacterial clock system rhythmicity

The characteristics of anKaiABC protein suggest that the circadian clock function of anKaiABC may differ from that of the modern

cyanobacterial circadian clock system. Since KaiABC produces sustainable KaiC phosphorylation rhythms in vitro (Nakajima et al, 2005), we conducted an in vitro oscillation assay and mixed KaiABC and anKaiABC proteins in multiple combinations to assess their circadian properties. As a control, KaiABC showed robust circadian rhythms with a period of ~22 h. In contrast, all of the other combinations presented no detectable rhythms, although anKaiB promoted KaiC phosphorylation at 30–40 h together with KaiA (Figs. 5A,B and EV3A,B), suggesting that anKaiB may

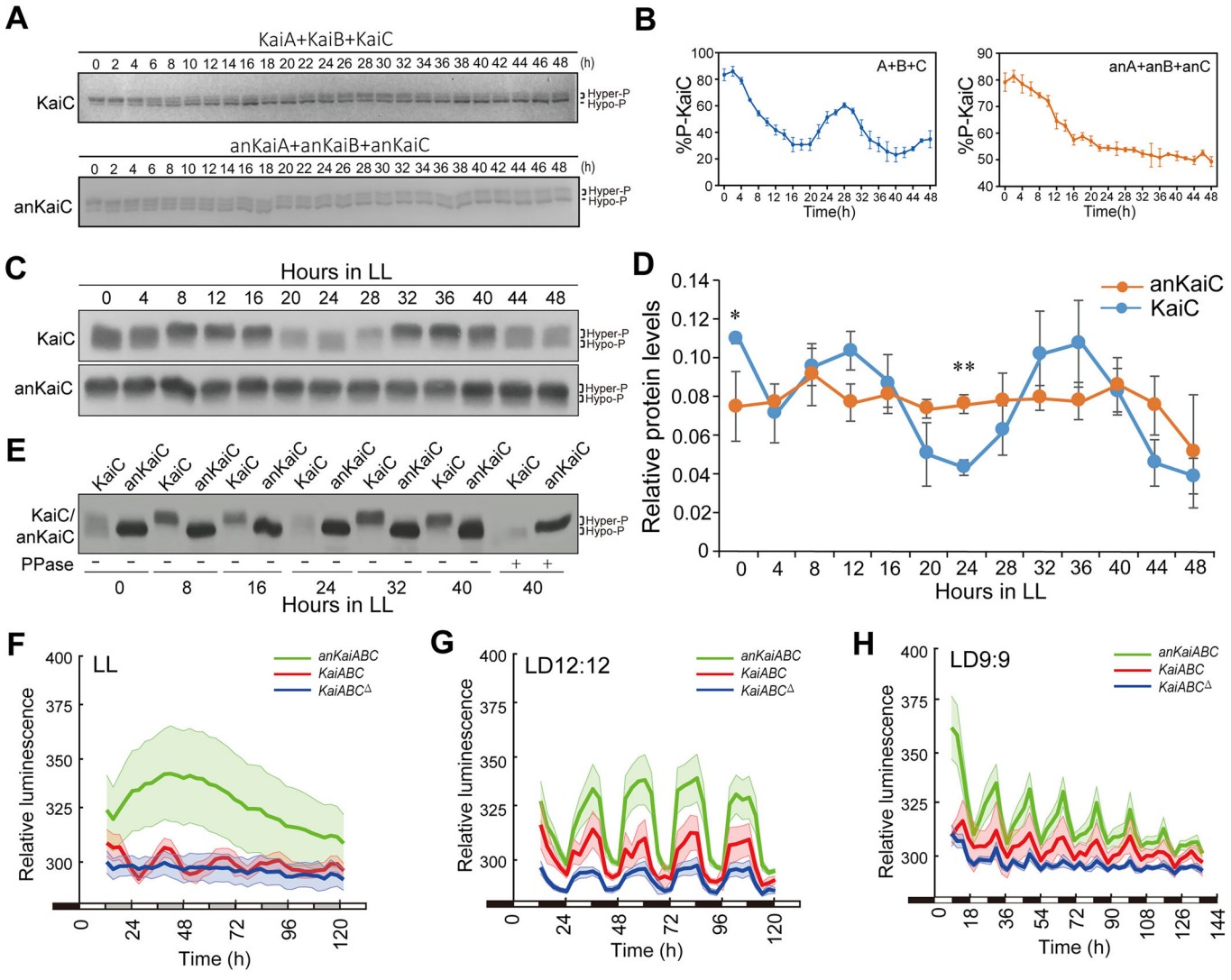

**Figure 5. Analysis of the circadian rhythmicity of the ancient circadian system.**

(A, B) Representative in vitro phosphorylation of anKaiC/KaiC proteins. Hyper-P denotes hyperphosphorylated proteins, and hypo-P denotes hypophosphorylated proteins, and the percentage of hyperphosphorylated KaiC was calculated. Representative results of Coomassie brilliant blue staining and curves of independent triplicates are shown. (C, D) Representative western blot results of KaiC/anKaiC in LL (C) and the statistical results (D). Data are means ± SE (*n* = 3, independent experiments). *P* values from left to right: *\*P* = 0.0298, *\*\*P* = 0.0025. (E) Comparison of KaiC/anKaiC phosphorylation in LL. *n* = 3. (F–H) Bioluminescence rhythms of *KaiABC*, *anKaiABC* and *KaiABC*Δ strains. *n* > 50. Data are means ± SD. Two-tailed unpaired Student's *t* test. Source data are available online for this figure.

possess a function relatively close to that of its modern counterpart.

To further validate whether the anKaiABC system can drive endogenous circadian rhythmicity, we transformed the *ankaiABC* genes into the *kaiABC* null strain (*kaiABC*Δ) of *Synechococcus sp.* PCC7942 with luciferase reporter driven by the *kaiBC* promoter (P*kaiBC::luxAB*) (Schopf and Packer, 1987). We then conducted Western blot analyses to compare the expression and phosphorylation of KaiC/anKaiC between the *kaiABC* and *ankaiABC* strains under different light/dark conditions. Under constant light (LL) condition, robust circadian rhythms of expression and phosphorylation were observed in the *kaiABC* strain but not in the *ankaiABC* strain (Fig. 5C–E).

Consistently, an in vivo bioluminescence assay revealed that the *KaiABC* strain, but not the *kaiABC*Δ and *ankaiABC* strains under LL

condition, presented overt circadian rhythms of bioluminescence (Fig. 5F). In contrast, all three strains presented overt bioluminescence rhythms under LD12:12 (Fig. 5G), and the *kaiABC* and *ankaiABC* strains but not the *kaiABC*Δ strain presented overt bioluminescence rhythms under LD9:9, which mimics the 18-h day-night period ~0.95 Ga ago (Fig. 5H). These data suggest that an ancient cyanobacterial circadian system can be entrained by environmental cues (light/dark cycles) despite the absence of endogenous rhythmicity.

## Adaptation of the ancient clock system to different light/dark cycles

The strains were monitored at 750 nm light absorption under LD9:9 and LD12:12 conditions to record their growth rate. LD9:9 simulates

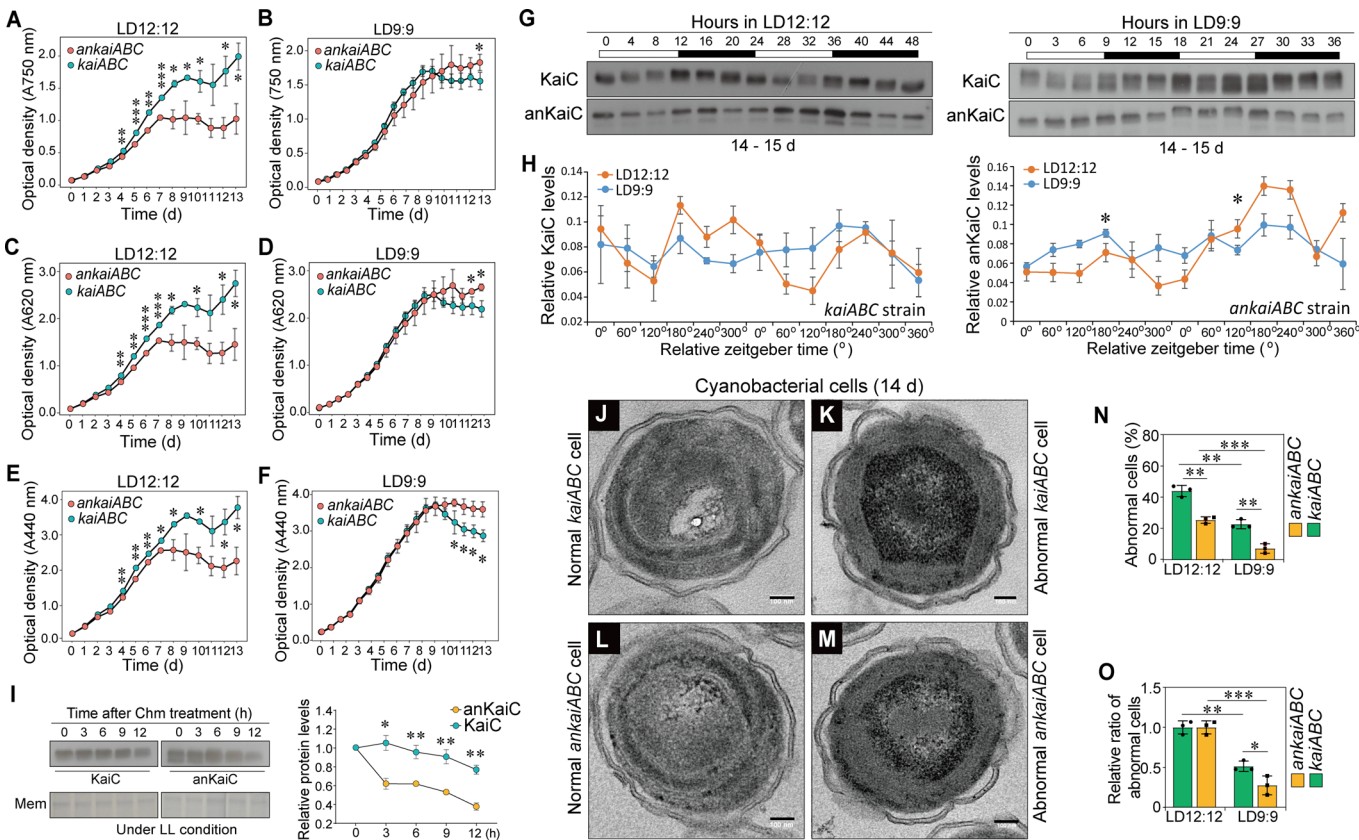

**Figure 6.  Contribution of ancient cyanobacterial circadian clock system to growth and adaptation.**

(A, B) Growth curves of *ankaiABC* and *kaiABC* strains measured by light absorption at 750 nm under LD9:9 and LD12:12 conditions. *P* values from left to right: \*\**P* = 0.0092, \*\*\**P* = 0.0001, \*\**P* = 0.0013, \*\*\**P* = 0.0005, \**P* = 0.0131, \**P* = 0.0461, \**P* = 0.0177, \**P* = 0.0151, \**P* = 0.0173, \**P* = 0.0393. (C, D) Phycobilin levels of *ankaiABC* and *kaiABC* strains measured by light absorption at 620 nm under LD9:9 and LD12:12 conditions. *P* values from left to right: \*\**P* = 0.0067, \*\**P* = 0.0027, \*\*\**P* = 0.0006, \*\*\**P* = 0.0008, \**P* = 0.0146, \**P* = 0.0152, \**P* = 0.0192, \**P* = 0.0196, \**P* = 0.0327, \**P* = 0.0112. (E, F) Chlorophyll a levels of *ankaiABC* and *kaiABC* strains measured by light absorption at 440 nm. *P* values from left to right: \*\**P* = 0.0066, \*\**P* = 0.0057, \*\**P* = 0.0023, \**P* = 0.0206, \**P* = 0.0492, \**P* = 0.0200, \**P* = 0.0209, \**P* = 0.0177, \**P* = 0.0431, \**P* = 0.0183, \**P* = 0.0174, \**P* = 0.0105. (G, H) Representative Western blot results of KaiC/anKaiC in LD12:12 and LD9:9 (G) and the statistical results (H). *P* values from left to right in (H): \**P* = 0.0357, \**P* = 0.0263. (I) Western blot results showing degradation of KaiC/anKaiC after chloramphenicol (Chm) treatment (800 µg/ml) under LL condition (left) and the statistical results (right). Membrane stained with amido black served as control. In the right panel: *P* values from left to right: \**P* = 0.0117, \*\**P* = 0.0099, \*\**P* = 0.0079, \*\**P* = 0.0030. (K–N) Observation of cells via TEM. The scale bars represent 100 nm. (N, O) Statistics of the results from (J–M). Scale bar is 100 nm. *P* values from left to right in (N): \*\**P* = 0.0015, \*\**P* = 0.0013, \*\*\**P* = 0.0010, \*\**P* = 0.0029; *P* values from left to right in (O): \*\**P* = 0.0013, \*\*\**P* = 0.0010, \**P* = 0.0378. Data are means ± SD. *n* = 3 (independent experiments) (A–F, H, I) and *n* = 3 (~100 cells) (N, O). Two-tailed unpaired Student's *t* test (A–F, H, I) and Unpaired Student's *t* test (N, O). Source data are available online for this figure.

the 18-h day length 0.95 Ga ago, whereas LD12:12 simulates the contemporary day length. The growth rate of the *kaiABC* strain before 4 d was similar to the *ankaiABC* strain under LD12:12, but afterwards, the growth rate of *kaiABC* strain was dramatically greater. In contrast, the growth rate of *ankaiABC* strain was slightly but significantly greater at 13 d under LD9:9 (Fig. 6A,B).

The absorption values at 620 nm and 440 nm light exposure wavelengths were also measured to quantify the relative density of the two photosynthetic pigments phycobilin and chlorophyll a, respectively (Nobel, 2020). The curves of phycocyanobilin and chlorophyll a exhibit similar patterns to those of the growth curves in each strain, and their levels were significantly greater in the *ankaiABC* strain under LD9:9 during the last several days. In contrast, the *kaiABC* strain showed significantly greater levels in these two photosynthetic pigments under LD12:12 (Fig. 6C–F). Moreover, the *kaiABC* strain showed significantly higher levels in

growth, phycobilin and chlorophyll a under additional short LD cycles including LD10.5:10.5 and LD7.5:7.5 (Fig. EV4A–F). Interestingly, the growth and photosynthetic pigment contents of the *kaiABC*^Δ strain were different from those of the *ankaiABC* strain under LD12:12 and LD9:9 conditions (Fig. EV5A–F).

We next conducted Western blots to compare the expression patterns of KaiC and anKaiC under LD12:12 and LD9:9 on 14 and 15 d, and the results revealed that the *kaiABC* strain lost its robust rhythmicity under LD9:9 compared with that under LD12:12 (Fig. 6G,H). We further treated the strains with chloramphenicol 800 µg/mL (Imai et al, 2004), to compare the degradation rates between anKaiC and KaiC under LL, and the results showed that anKaiC was less stable than KaiC (Fig. 6I).

The transmission electron microscopy (TEM) results revealed that some cyanobacterial cells presented abnormal phenotypes of

crenated cell walls and dissembled phycobilisomes under LD12:12 and LD9:9 conditions (Fig. 6J–M). The abnormal cell ratios in both the *ankaiABC* and control strains were significantly greater at 14 d than those at 1 d, demonstrating the depletion of nutrition due to excessive growth (Fig. 6N). Interestingly, the increase in the ratio of abnormal cells was significantly lower in the *ankaiABC* strain than in the control under LD9:9 condition (Fig. 6O).

Moreover, we conducted competition experiments between these two strains grown for 14 d under LD12:12 and LD9:9 light-dark conditions (Fig. 7A). As illustrated in Fig. 7B,C, *kaiABC* outcompeted *ankaiABC* under LD12:12 condition while *ankaiABC* outcompeted under LD9:9. These data demonstrate that the *ankaiABC* strain may have adapted to the shorter day-night cycles with a ~18-h period of day length occurring ~0.95 Ga ago.

# Discussion

The palaeontologic evidence from the daily or monthly growth rates of corals and shells demonstrates that the rotation of Earth is slowing and that the day length is increasing as a consequence (Wells, 1963). However, fossils cannot tell the dynamic story of the circadian clock at the molecular level. The reconstruction of the ancient cyanobacterial clock system may serve as an evolutionary tool for understanding the features and functions of the ancient circadian system (Hochberg and Thornton, 2017). In this study, we reconstituted the ancient cyanobacterial circadian clock system of KaiABC proteins, and the entrainment observed under LD12:12 and LD9:9 conditions implies that the anKaiABC protein can be integrated with circadian inputs and outputs pathways in contemporary cyanobacterial cells (Fig. 5G,H).

KaiC is the core component in the KaiABC system which can form hexamers; KaiA and KaiB regulate the phosphorylation of KaiC (Kawamoto et al, 2020; Nishiwaki et al, 2000; Nishiwaki and Kondo, 2012). Likewise, anKaiC also forms hexamers as revealed by cryo-EM (Fig. 2A), however, the cryo-EM and SANS results revealed differences in surface hydrophobicity, conformation, and interactions between the ancient and contemporary circadian components (Fig. 2B–J). In vitro biochemical assays revealed that anKaiA can promote the phosphorylation of KaiC despite its activity being lower than that of KaiA. In contrast, anKaiA did not affect anKaiC phosphorylating (Fig. 3B), which may be due to the dramatic differences in structure between anKaiA and KaiA, anKaiC and KaiC, and the different interactions between KaiAC and anKaiAC (Fig. 2H–J). Consistently, some species with a primitive circadian system lacking kaiA cannot generate self-sustainable oscillation, suggesting the critical role of KaiA in maintaining the function of the cyanobacterial circadian clock (Hochberg and Thornton, 2017; Terauchi et al, 2007).

Mutation of KaiC-V118A showed no impact on KaiC dephosphorylation which may be explained by that this mutation does not affect function of the highly flexible structure of B-loop (Figs. 2C and 4A–C). These results suggest that the evolution of anKaiC-A122 is not critical for promoting KaiC dephosphorylation, whereas it is important for maintaining KaiC phosphorylation through the mechanisms remain unknown (Fig. 4B,C).

The ATPase activity of anKaiC is much lower than that of contemporary KaiC (Fig. 3E), which can be at least in part

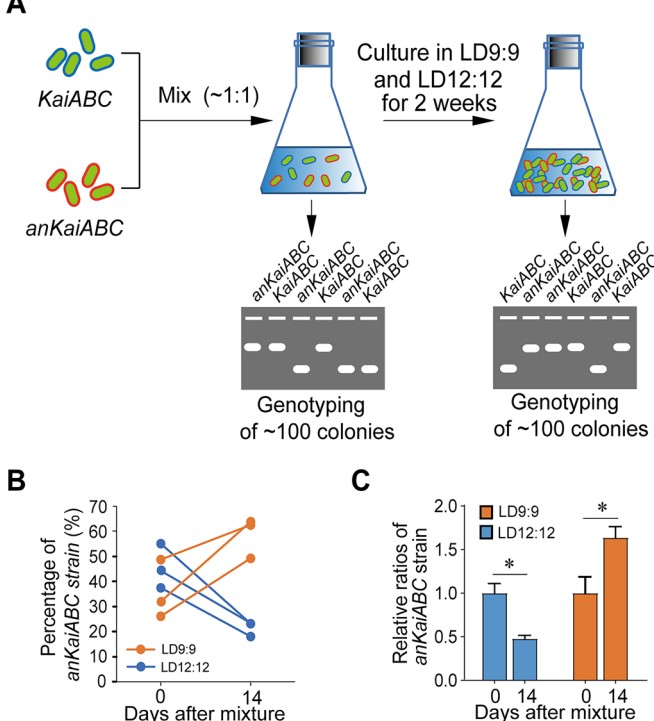

**Figure 7. Growth of ancient and modern cyanobacterial strains in competition with each other under different LD conditions.**

(A) Illustration of the competition experiment. Two initial strains of interest are mixed (~1:1) and after 14 d single colonies derived from the culture are subjected to genotyping to determine the ratio of the two strains. PCR of ~100 colonies was conducted to screen and confirm their genotypes. The strains are denoted in different colours. (B) Competition results between *kaiABC* and *ankaiABC* strains under LD9:9 and LD12:12 conditions. Triplicate results are shown, and in each experiment, approximately 100 colonies were detected. (C) Relative competitive strength of the two strains as a function of the ratio between the final and initial proportions. Data are means ± SE (*n* = 3, ~100 colonies). Two-tailed unpaired Student's *t* test. *P* values from left to right: *$P = 0.0113$, *$P = 0.0496$. Source data are available online for this figure.

attributed to the residues T233/N244 spatially neighbouring to ATP molecules (Figs. 2E–G and 4F). The C-terminal part of A-loop in anKaiC has several amino acid residues different from those in KaiC (Figs. 1F and EV1F), suggesting this region may also have a different structure from anKaiC which accounts for the lower level of anKaiC phosphorylation through affecting KaiB-KaiC interaction. The synergetic alteration in critical residues, for instance, T233 and N244, may contribute importantly to structural and functional evolution of KaiC.

The combination of anKaiA-anKaiB-anKaiC displayed no circadian rhythm in terms of anKaiC phosphorylation, and the phosphorylated level of anKaiC was consistently low (Fig. 5A,B). Similarly, in the 24-h cycles, the *kaiABC* strain displayed circadian rhythms at the KaiC protein level, whereas the *ankaiABC* strain was arrhythmic (Fig. 5C–E). Together, these data demonstrate that, compared with the modern KaiABC system, the ancient KaiABC system already had a partial circadian clock function; however, it was still too evolutionarily immature to maintain self-sustainable circadian oscillation. A

sustainable circadian system may have developed sometime after 0.95 Ga ago in cyanobacteria.

In different bacteria, an hourglass-type clock can also coordinate physiological processes with the cycling environment in some cyanobacterial species harbouring no intact KaiABC system (Hochberg and Thornton, 2017; Chew et al, 2018; Jabbur and Johnson, 2022; Ma et al, 2016; Pitsawong et al, 2023). Similarly, despite the lack of an endogenous circadian clock in the *ankaiABC* strain, we showed via bioluminescence experiments that this strain can be entrained under LD12:12 and LD9:9 conditions (Fig. 5G,H), suggesting that the ancient anKaiABC system may function as an hourglass-type clock.

*Cyanobacteria* expressing ancient circadian genes presented a significantly increased growth rate and greater content of photosynthetic pigments after exhaustive growth under LD9:9 at the last stage (Figs. 6A–F and EV4A–F), and competitive strength in shorter day-night cycles (Fig. 7). Furthermore, the *ankaiABC* strain showed no overt entrained bioluminescence rhythmicity under LD9:9, and presented different growth and photosynthetic pigment levels compared with the *kaiABC*$^\Delta$ strain (Figs. 5H and EV5A–F), suggesting that the ancient proto-circadian system already possessed certain functions in modulating diurnal rhythms and adaptations to the environment despite its lack of self-sustainability. As the content of atmospheric $O_2$ ~0.95 Ga ago was less than 1/10 of the current level (Poulton, 2017), it is also likely that the ancestral circadian system may have conferred adaptability to cyanobacteria, enabling them to grow at a relatively higher rate in a low oxygen environment.

The ATPase activity of KaiC is critical for its circadian function and the rate-limiting reaction for its circadian period length, which is very slow and is temperature compensated (Axmann et al, 2009; Terauchi et al, 2007; Abe et al, 2015). However, the anKaiABC system does not exhibit intrinsic rhythmicity with a long period despite the lower ATPase activity of anKaiC (Fig. 3E). The correlation between ATPase activity and circadian period may not apply to the primordial anKaiABC system which acts in an hourglass-like fashion. Instead, the higher turnover rate of anKaiC may contribute to its adaptation to the short day length (Fig. 6I).

Together, these data suggest that the ancient cyanobacterial circadian components already had certain functions in regulating growth or metabolism despite the lack of endogenous oscillation. The ancient proto-circadian clock system may be beneficial for cyanobacterial survival in the 18-h daily cycle ~0.95 Ga ago.

# Methods

### Reagents and tools table

| Reagent/resource | Reference or source | Identifier or catalog number |
| --- | --- | --- |
| **Experimental models** | | |
| *Synechococcus sp.* PCC7942 *kaiABC*$^\Delta$ | Obtained from Johnson CH Lab | N/A |
| *Synechococcus sp.* PCC7942 strain harbouring a *luxAB* reporter (P*kaiBC*::*luxAB*) | Obtained from Johnson CH Lab | N/A |

| Reagent/resource | Reference or source | Identifier or catalog number |
| --- | --- | --- |
| *Synechococcus sp.* PCC7942 *kaiABC*$^\Delta$: *kaiABC* | This study | N/A |
| *Synechococcus sp.* PCC7942 *kaiABC*$^\Delta$: *ankaiABC* | This study | N/A |
| **PCR primers** | | |
| Forward primer for construction of cyanobacterial transformation plasmids *pKai-TR-ankaiABC* and *pKai-TR-ankaiBC* (V-A-homo-F) | This study | 5'-cagcggcggagg aactctttga-3' |
| Reverse primer for construction of cyanobacterial transformation plasmids *pKai-TR-ankaiABC* and *pKai-TR-ankaiBC* (V-A-homo-R) | This study | 5'-tcaaagagtt cctccgccgctg-3' |
| Forward primer for construction of cyanobacterial transformation plasmid *pKai-TR-ankaiABC* (V-A-F) | This study | 5'-aaaggtaaaggagg tcttaagctcgg-3' |
| Reverse primer for construction of cyanobacterial transformation plasmids *pKai-TR-ankaiABC* and pKai-TR-ankaiBC (V-A-R) | This study | 5'-gcagtcgctcctgtc aggac-3' |
| Forward primer for construction of cyanobacterial transformation plasmid *pKai-TR-ankaiABC* (V-B-F) | This study | 5'-ctgtcgttaactg ctttgttggtact-3' |
| Reverse primer for construction of cyanobacterial transformation plasmid *pKai-TR-ankaiABC* (V-B-R) | This study | 5'-acgcagatcaacg gggtagca-3' |
| Forward primer for construction of cyanobacterial transformation plasmid *pKai-TR-ankaiABC* (V-C10-F) | This study | 5'-agctttatgcttgt aaaccgttttgt-3' |
| Reverse primer for construction of cyanobacterial transformation plasmid *pKai-TR-ankaiABC* (V-C10-R) | This study | 5'-cttaaaagagggtga agtcaggtagt-3' |
| Forward primer for construction of cyanobacterial transformation plasmid *pKai-TR-ankaiABC* (An-A10-F) | This study | 5'-gtcctgacaggagcgactg catgttatctgcct taacgatct-3' |
| Reverse primer for construction of cyanobacterial transformation plasmid *pKai-TR-ankaiABC* (An-A10-R) | This study | 5'-ttaagacctcctttacctttt taagaagaaa cctcagagtc-3' |
| Forward primer for construction of cyanobacterial transformation plasmid *pKai-TR-ankaiABC* (An-B10-F) | This study | 5'-gctaccccgttg atctgcgtat gagcccgctgcgc aaaaccta-3' |
| Reverse primer for construction of cyanobacterial transformation plasmid *pKai-TR-ankaiABC* (An-B10-R) | This study | 5'- aacaaagcagttaacgac agttacggatcgct atccggc-3' |
| Forward primer for construction of cyanobacterial transformation plasmid *pKai-TR-ankaiABC* (An-C10-F) | This study | 5'-tgacttcaccc tcttttaagat gaccagcctggcggaacat-3' |
| Reverse primer for construction of cyanobacterial transformation plasmid *pKai-TR-ankaiABC* (An-C10-R) | This study | 5'-cggtttacaagcataaagc tttagctttccggt tcttttttcc-3' |
| Forword primer for construction of cyanobacterial transformation plasmid *pKai-TR-ankaiBC* (anBC-F) | This study | 5'-gacaggagcga ctgcaaaggtaa aggaggtcttaagctcgg-3' |

| Reagent/resource | Reference or source | Identifier or catalog number |
|---|---|---|
| Reverse primer of *KaiA* for genotyping (cpWt1-F) | This study | 5′-tgaaccagc caaagaaca-3′ |
| Reverse primer of *KaiA* for genotyping (cpWt1-R) | This study | 5′-ggaacatcggc aaagaaa-3′ |
| *anKaiB+anKaiC* for genotyping (anAn1-F) | This study | 5′-gaactgctgc tggatgat-3′ |
| *anKaiB+anKaiC* for genotyping (anAn1-R) | This study | 5′-gatccacca gtttctggca-3′ |
| **Recombinant DNA** | | |
| *pGEX-6P-1* | Addgene | Cat#: 27-4597-01 |
| *pGEX-6P-1-kaiC* | This study | N/A |
| *pGEX-6P-1-kaiA* | This study | N/A |
| *pGEX-6P-1-kaiB* | This study | N/A |
| *pGEX-6P-1-anKaiC* | This study | N/A |
| *pLou3-ankaiA* | This study | N/A |
| *pGEX-6P-1-ankaiB* | This study | N/A |
| *pGEX-6P-1-kaiC-V118A* | This study | N/A |
| *pGEX-6P-1-kaiC-S229T* | This study | N/A |
| *pGEX-6P-1-kaiC-T240N* | This study | N/A |
| *pGEX-6P-1-KaiC-S229T-T240N* | This study | N/A |
| *pGEX-6P-1-psp* | This study | N/A |
| *pRK793* | Addgene | Cat#: 8827 |
| *pKai-TR-kaiABC* | Obtained from Johnson CH Lab | Nishimura et al, 2002 |
| *pKai-TR-ankaiABC* | This study | N/A |
| *pKai-TR-ankaiBC* | This study | N/A |
| **Antibodies** | | |
| Rabbit anti-KaiC 1:5000 for immunoblotting | This study | N/A |
| Goat Anti-Rabbit IgG (H + L)-HRP Conjugate | Bio-Rad | Cat#:170-6515 |
| **Chemicals, enzymes and other reagents** | | |
| TransStart® Taq DNA Polymerase | TransGen Biotech | AP141-13 |
| FastPure EndoFree Plasmid Midi Kit | Vazyme | DC205-01 |
| BG11 | Hopebio | HB8793 |
| Agar Powder | Hopebio | HB8274 |
| Chloramphenicol | Sangon Biotech | A100230 |
| Spectinomycin | Sangon Biotech | A600901 |
| Kanamycin | Sangon Biotech | A600286 |
| Agar | Sangon Biotech | A505255 |
| Tryptone | Oxoid | LP0042B |
| Yeast extract | Oxoid | LP0021 |
| NaCl | Sangon Biotech | A501218 |

| Reagent/resource | Reference or source | Identifier or catalog number |
|---|---|---|
| Tris | Sangon Biotech | A600194 |
| IPTG Dioxane Free | Sangon Biotech | A600168 |
| *Bam* HI | New England Biolabs | R3136V |
| *Sac* I | New England Biolabs | R3156S |
| Glutathione reduced | Sangon Biotech | A600229 |
| Glutathione Sepharose 4B | Coolaber | CS20421 |
| ATP | Sigma-Aldrich | A1852 |
| ADP | Sigma-Aldrich | A2754 |
| **Software** | | |
| SnapGene | SnapGene Software | https://www.snapgene.com/ |
| ImageJ | National Institutes of Health | https://imagej.nih.gov/ij/ |
| R software | R Foundation Statutes | https://www.r-project.org/ |
| GraphPad Prism 8 | Graphpad Software | https://graphpad.com |
| CRYSON | Svergun et al, 1998 | https://www.embl-hamburg.de/biosaxs/cryson.html |
| SasView | SasView Software | https://www.sasview.org/ |
| **Other** | | |
| anKaiC structure | This study | PDB:8JON |
| KaiC structure | Swan et al, 2022 | PDB:7S67 |

## Cyanobacterial strains

The *Synechococcus sp*. PCC7942 strain harbouring a *luxAB* reporter (*pkaiBC::luxAB*) under the control of the *kaiBC* promoter at a neutral site, and the *Synechococcus sp*. PCC7942 *kaiABC*$^\Delta$ strain in which the locus of *kaiABC* is replaced by the kanamycin resistance gene (*kmr*) were gifts from Carl H. Johnson lab (Vanderbilt University, USA) (Nishimura et al, 2002). To prepare strains expressing *ankaiABC* genes, these genes were cloned and inserted into the *pkai-TR* plasmid which contains homologous flanking arms (526 nt upstream of *kaiA* and 288 nt downstream *kaiC*). The plasmids of *pkai-TR-kaiABC* and *pkai-TR-ankaiABC* were transformed into the *kaiABC*$^\Delta$ strain and recombined into the locus of *kmr* using the natural transformation method (Edgar et al, 2012). In the transformed strains, *ankaiA* was driven by the *kaiA* promoter and *ankaiBC* was driven by the *kaiBC* promoter, respectively. Cyanobacteria were cultured under the indicated conditions with an intensity of ~50 µE m$^{-2}$ s$^{-1}$ of light in a light- and temperature-controllable incubator (Percival, USA) (Ouyang et al, 1998).

## Phylogenetic analysis of the ancestral sequences of cyanobacterial clock genes

We reconstructed the protein sequences of the ancestral KaiA, KaiB, and KaiC for the node at 0.95 Ga ago (Fig. 1A–C). The phylogenetic trees were constructed on the basis of the 16S rRNA sequences and the divergence times along with the cyanobacterial phylogeny were estimated under Bayesian relaxed molecular clocks (Schirrmeister et al, 2011; Schirrmeister et al, 2013). The amino acid sequences of KaiA, KaiB, and KaiC from the *Cyanobacteria* AC group, ranging from Prochlorococcus marinus to *Synechococcus sp* and *S. elongatus* PCC7942, were used for ancestral sequence reconstruction (Schirrmeister et al, 2013). These sequences were aligned with the program MUSCLE. The PAML package was used to infer the posterior amino acid probability per site under the LG model of protein evolution, with the phylogenetic tree constructed based on the 16S rRNA sequences (Yang, 2007).

## Prokaryotic protein expression and purification

The *kaiABC* gene cassettes were obtained from PCR amplification using the genomic DNA of *Synechococcus sp.* PCC7942 as templates, and *ankaiABC* genes were synthesized by Shenggong Co. Ltd. (Shanghai, China). Synthesis of the ancestral genes of *kaiA*, *kaiB*, and *kaiC* were based on the *E. coli* codon usage table. KaiC mutations of V118A, S229T, T240N, and the double mutations S229T/T240N were generated through site directed mutagenesis.

The cassettes of *kaiABC*, *ankaiBC*, and *kaiC* mutant genes were cloned and inserted into the *pGEX-6p-1* vector, respectively, and *ankaiA* was cloned into the *plou3* vector. The plasmids were transformed into *E. coli* BL21 and the target proteins were purified as reported previously (Dvornyk, 2006). For each protein, 1 L of culture was grown at 37 °C to an OD600 of ~0.6, at which point it was cooled to 25 °C and induced overnight with 100 µM IPTG. The cells were subsequently harvested, pelleted, and frozen at −80 °C. anKaiABC and kaiABC proteins were purified with GSTrap column (GE Healthcare, USA), Ni-NTA column (Qiagen, Germany) and Resource Q anion exchange column (GE Healthcare, USA). The protein concentration was determined by the Bradford method. The purity of anKaiA was >90% as determined by SDS/PAGE.

## Cryo-EM sample preparation, data acquisition and processing, and model building

Three microliters of freshly purified anKaiC were applied to a glow discharged R1.2/1.3 holey copper grid (Quantifoil Micro Tools GmbH, USA). The grid was then blotted and plunged in precooled liquid ethane using a Vitrobot Mark IV (Thermo Fisher Scientific, USA). The cryo-EM data of anKaiC were collected under a Titan Krios G4 300 kV electron microscope (Thermo Fisher Scientific) equipped with a Falcon4 camera (Thermo Fisher Scientific, USA). Data collection was performed using *EPU* software (Thermo Fisher Scientific, USA) with defocus values ranging from 1.0 to 3.0 µm. Video frames were recorded in electron-event representation (EER) mode under a dose rate of 16.2 $e^-$/Å$^2$/s, giving a total dose of 50 $e^-$/Å$^2$. The nominal magnification was 165,000×, giving a calibrated pixel size of 0.71 Å. Beam-induced motion correction and dose weighting were performed using Relion3 (Zivanov et al, 2018). Contrast transfer function parameters were estimated using

CTFFIND4. Protein particles were automatically picked by the BoxNet convolutional neural network in the program Warp. Rounds of reference-free 2D classification and 3D classification were performed using Relion3 to select clear and homogeneous particles. Higher-order aberrations, anisotropic magnification, and the x defocus value of each particle were refined, and the particles were polished. Then, the polished particles were 3D refined and generated the final reconstruction using Relion3. The resolution was generated using the gold standard FSC = 0.143 criterion. The maps were sharpened using Phenix. The atomic model of anKaiC was initially built by AlphaFold2 and then manually tuned in COOT based on the electron density map. The model was refined and validated in Phenix. Refinement and validation statistics are summarized in Table EV1. All cryo-EM structure figures were prepared in UCSF ChimeraX.

## In vitro measurement of the autokinase and autophosphatase activities of KaiC/anKaiC

KaiC function was measured as previously described with some modifications (Dvornyk, 2006). To assess its autophosphatase activity, initially anKaiC protein (0.2 µg/µL) was incubated in reaction buffer (50 mM Tris-HCl, pH 8.0, 150 mM NaCl, 1 mM DTT, 5 mM MgCl$_2$, 1 mM ATP) at 30 °C. Aliquots of 5 µl reaction mixtures were collected every 2 h and stored at −20 °C. To assess the autokinase activity of anKaiC, after 15-h incubation the reaction mixture was transferred to 0 °C and 5 µl aliquots were sampled until 27 h after initiation (Snijder et al, 2014). All aliquots were resolved by electrophoresis on 10% separation gel at 175 V for 80 min. The gel was dyed with Coomassie brilliant blue G-250. NIH ImageJ software (version 1.51) was used for densitometric analysis.

## Small angle neutron scattering analysis of cyanobacterial clock proteins

KaiABC and anKaiABC proteins were prokaryotically expressed and purified as described above, and the purified proteins were subjected to ultrafiltration. Sedimentation of KaiB protein occurred during ultrafiltration, therefore, it was precluded for SANS analysis. The protein quantities and concentrations for SANS experiments were: KaiA, 1.67 mg (8.8 µg/µL); anKaiA, 1.68 mg (8.4 µg/µL); KaiC, 1.73 mg (9.6 µg/µL); anKaiC 1.67 mg (9.6 µg/µL). The ratio of KaiAC/anKaiAC proteins was KaiA/anKaiA:KaiC/anKaiC=1:4 (Nakajima et al, 2005). The solvent water in all solutions were substituted with D$_2$O, including the solution for KaiC (1 M Tris-HCl, 5 M NaCl, 1 M MgCl$_2$, 0.5 M ATP, 1 M DTT, pH 8.0), and the solution for KaiA (1 M Tris-HCl, 5 M NaCl, 1 M DTT, pH 8.0). The proteins were incubated and measured at 30 °C.

The SANS experiments were carried out on the multi-slit very small angle neutron scattering (MS-VSANS) instrument at the China Spallation Neutron Source (CSNS, https://user.csns.ihep.ac.cn/). The samples were contained within quartz banjo cells with a sample thickness of 1 mm. Each sample, along with the buffer, underwent measurement sessions lasting for 1–2 h. The measurements were conducted with a 9.92 m collimation length and neutron wavelengths ranging from 2.2 to 6.7 Å (Zuo et al 2024). The absolute intensities were derived utilizing the direct beam method, incorporating corrections for solid angle and background subtraction. Theoretical scattering profiles of the

proteins were computed using CRYSON (Svergun et al 1998), accounting for instrumental resolution effects.

The data of KaiA and anKaiA were fitted with the poly_-excl_volume model with the SasView software (version 5.0.6, https://www.sasview.org/docs/old_docs/5.0.6/index.html). There was no suitable model for fitting the KaiC/anKaiC data in SasView, therefore, a dumbbell model was constructed with the Python scripts based on the cryo-EM data. The parameters of dumbbell model were modified to obtain the best fit of KaiC and anKaiC with the particle swarm algorithm. The dumbbell model consists with two identical ellipsoids with a cylindrical hole go through the two ellipsoids (Fig. EV2A–C). The parameters for anKaiC modelling were: Ellipsoids center distance, 53 Å; Ellipsoid radii a, b, c, 35.4 Å, 45.5 Å. 45.5 Å; Cylinder hole radius, 17.4 Å. The parameters for KaiC modelling were: Ellipsoids center distance, 50 Å; Ellipsoid radii a, b, c, 32.4 Å, 44.5 Å. 44.5 Å; Cylinder hole radius, 16.8 Å.

## In vitro measurement of KaiA/anKaiA functions in promoting KaiC phosphorylation

The KaiC/anKaic proteins were incubated in the solution with 0.2 μg/μL KaiC, 1 mM ATP, 5 mM MgCl$_2$, 150 mM NaCl, 1 mM DTT and 50 mM Tris-HCl buffer (pH 8.0) at 30 °C, to induce hypophosphorylation of KaiC/anKaiC. After 15 h of incubation, KaiA or anKaiA proteins were added to the system (final concentration: 0.05 μg/μL) and incubated for another 12 h. Five microliter aliquots were taken out from each system every 3 h. All aliquots were resolved by electrophoresis on a 10% separation gel at 175 V for 80 min. The gel was dyed with Coomassie brilliant blue G-250. NIH ImageJ software (version 1.51) was used for densitometric analysis.

## In vitro measurement of KaiB/anKaiB functions in repressing KaiC phosphorylation

The KaiA/anKaiA (final concentration: 0.05 μg/μL) and KaiC/anKaiC (0.2 μg/μL KaiC) were added to the solution with 1 mM ATP, 5 mM MgCl$_2$, 150 mM NaCl, 1 mM DTT and 50 mM Tris-HCl buffer (pH 8.0) and incubated at 30 °C for 12 h to induce hyperphosphorylation of KaiC or anKaiC. KaiB/anKaiB proteins were added to the solution (final concentration: 0.05 μg/μL) and incubated for 15 h. Five microliter aliquots were taken out from each system every 3 h. All aliquots were resolved by electrophoresis on a 10% separation gel at 175 V for 80 min. The gel was dyed with Coomassie brilliant blue G-250. NIH ImageJ software (version 1.51) was used for densitometric analysis.

## In vitro ATPase assay of KaiC

The ATPase activity of purified KaiC was measured through HPLC method (Terauchi et al, 2007). Briefly, the purified KaiC protein (0.5 pmol hexamer/μL) was incubated at 25 °C in buffer [20 mM Tris·HCl (pH 8.0), 150 mM NaCl, and 5 mM MgCl$_2$]. KaiC/anKaiC proteins were incubated with 1 mM ATP for the periods indicated in the presence (closed, solid line) or absence (open, dashed line) of KaiA (3.0 pmol/μL). The HPLC system (1290 Infinity, Agilent Technology, America) was equipped with a DAD detector; and a Perfect T3 (4.6 × 250 mm, 5 μm) column (Micropulite, China) as described. ATP and ADP in the reaction mixture were separated on

the column at 40 °C at a 1.0 ml/min flow rate. The detection wavelength was set at 260 nm. The mobile phase was composed of 50 mM NaH$_2$PO$_4$ buffer (pH 7.0) and 50% (vol/vol) acetonitrile. The ATPase activity was evaluated as a function of the amount of ADP produced (Terauchi et al, 2007).

## In vitro reconstitution of KaiC phosphorylation oscillation

The reconstitution of KaiC phosphorylation rhythm in vitro was performed as previously described (Nakajima et al, 2005). Briefly, KaiC (0.2 μg/μl) was incubated with KaiA (0.05 μg/μl) and KaiB (0.05 μg/μl) in reaction buffer (50 mM Tris-HCl, pH 8.0, 150 mM NaCl, 1 mM DTT, 5 mM MgCl$_2$, 1 mM ATP) at 30 °C. Aliquots of the 5 μl reaction mixtures were collected every 2 h, and the reaction was stopped by the addition of SDS loading buffer. Samples were subjected to SDS-PAGE on 10% gels followed by Coomassie Brilliant Blue staining. NIH ImageJ software (version 1.51) was used for densitometric analysis.

## Generation of KaiC multi-clone antibody and Western blot analysis

KaiC protein was expressed in E. coli, extracted from inclusion bodies, and purified with a column containing Ni-NTA agarose (Qiagen, Germany). The resultant purity of KaiC protein was >90%, which was used to immunize a rabbit. The serum containing antibodies against KaiC was extracted and validated. Synechococcus cells were resuspended in 100–200 μL of lysis buffer (25 mM Tris pH 8.0, 0.5 mM EDTA, 1 mM DTT and protease inhibitor cocktail), and the total protein was isolated and Western blot experiments were performed.

## Transformation of cyanobacteria

Transformation of cyanobacteria was performed by following the previously published methods (Nishiwaki and Kondo, 2012). Briefly, 5–10 ml of cyanobacteria cells were centrifuged at 1620 g for 5 min and washed in 10 ml of 10 mM NaCl. Then the cells were resuspended in 0.3 ml of BG11 medium. For transformation, 1 μl of miniprep plasmid (0.1–0.3 μg) was added to the cells and incubated at 30 °C overnight with gentle shaking in darkness. The transformed cells were grown on BG11-agar solid medium containing antibiotics, including chloramphenicol and spectinomycin. The transformed cells were grown on plates at 30 °C under LL condition until the formation of the transformed colonies. Single colonies were picked and grown in BG11 liquid media under LL. When the culture became green, PCR and sequencing were conducted for validation.

## Bioluminescence monitoring and associated analysis

S. elongatus strains expressing PkaiBC-luxAB were grown on Petri dishes with BG11 media at 30 °C. The strains were inoculated onto BG-11 solid medium in 90-mm plates and the bioluminescence intensity of the colonies was monitored by a CCD camera (iKon-M 934, Andor, Northern Ireland). The bacterial colonies were approximately 0.5–1.0 mm in diameter and contained 100–1000 cells. Prior to monitoring, the plates were exposed to DD for 12 h to synchronise the circadian rhythms. Circadian rhythms of LUC

activity were captured using a back-illuminated CCD sensor from e2v (CCD47-40) and normalized to the mean value over the time series. Fast Fourier transform-nonlinear least squares analysis of circadian parameters was conducted on a data window of ZT24-120. The bioluminescence activity of LUC fusion proteins was measured on a Packard TopCount™ luminometer and used as a read-out of the state in LD12:12 and LL conditions, and the light intensity was 50 μE m$^{-2}$ s$^{-1}$ during the light regimes. N-decanal (3% v/v, dissolved in canola oil) was used as the substrate of luciferase and the final concentration was 3% v/v (Mackey et al, 2007). The bioluminescence data were analysed with R-4.2.1).

### Transmission electron microscopy (TEM) sample preparation and results analysis

*Synechococcus* cells were harvested by centrifugation at $1500 \times g$, fixed in 2.5% glutaraldehyde and 2% paraformaldehyde prepared in 0.1 M PBS (pH 7.2) at 4 °C for 4 h, and treated with 2% osmium tetroxide at room temperature overnight. Thereafter, the samples were gradient dehydrated and embedded in Spurr resin (Sigma-Aldrich Co. Ltd., USA). Ultrathin sections were cut and stained with aqueous uranyl acetate and lead citrate. The grids were examined with a JEM1400Flash electron microscope (JOEL Co. Ltd., Japan) operating at an acceleration voltage of 120 kV. For each replicate, the phenotypes of 100 randomly selected cells were observed and counted. The thickness of ultrathin sections was 50–100 nm.

### Competition experiment with cyanobacterial strains

The competition experiments were conducted as previously described (Ouyang et al, 1998). In the competition experiments, cyanobacteria were cultured on BG-11 solid or liquid medium under the indicated conditions with fluorescence lamps with a light intensity of 90–100 μE m$^{-2}$ s$^{-1}$. To calculate the ratios of different strains in the culture mixture, 50 μL of the mixture was taken out and grown on plates to obtain single colonies. Genomic DNA samples were extracted from >100 single colonies and subjected to PCR amplification and electrophoresis for genotyping to determine the composition of populations. The sequences of the PCR primers are listed in the Reagents and Tools Table.

### Statistical analyses

Statistical analyses were generally performed using R version 4.2.1 or with the software GraphPad Prism 8.0.2. The TEM results were analysed with unpaired $t$ test, and others analysed with two-tailed unpaired Student's $t$ test. Data are plotted as mean ± standard deviation (SD) or mean ± standard error (SE) as indicated. $P$ values obtained are indicated in the respective figure legends.

## Data availability

The cryo-EM maps of anKaiC were deposited in the Electron Microscopy Data Bank (EMDB) under the accession code EMD-36461. The atomic coordinates were deposited in the RCSB Protein Data Bank (PDB) under the accession code 8JON. The source data of this paper are collected in the following database record: https://www.ebi.ac.uk/biostudies/studies/S-BSST1691.

The source data of this paper are collected in the following database record: biostudies:S-SCDT-10_1038-S44318-025-00425-0.

## Peer review information

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

## Acknowledgements

We thank Y Xu and CH Johnson (Vanderbilt University) for providing strains and technical help; Y Liu (UT Southwestern Medical Center) and Y Li (Hongkong University) for helpful discussion; D Chen and W Zhuo (Sun Yat-sen University) for technical support. This work was supported by grants to JG, including the National Key R&D Program of China (2022YFA1604504), the Lin Gang Laboratory & National Key Laboratory of Human Factors Engineering of the Astronaut Center of China Joint Grant (LG-TKN-202203-01), the Experiments for Space Exploration Program and the Qian Xuesen Laboratory, China Academy of Space Technology (TKTSPY-2020-04-21), the Open Fund of the National Key Laboratory of Human Factors Engineering in the Astronaut Center of China (SYFD062008K), the National Natural Science Foundation of China (32171162), the Space Medical Experiment Program (HYZHXMN01003),

and the Zhongyuan Science and Technology Innovation Leadership Talent Project to X.X. (234200510023).

## Author contributions

**Silin Li**: Formal analysis; Validation; Investigation. **Zengxuan Zhou**: Formal analysis; Validation; Investigation. **Yufeng Wan**: Formal analysis; Validation; Investigation. **Xudong Jia**: Formal analysis; Validation; Investigation. **Peiliang Wang**: Formal analysis; Investigation. **Yu Wang**: Formal analysis; Investigation. **Taisen Zuo**: Formal analysis; Validation; Investigation; Visualization. **He Cheng**: Formal analysis; Validation; Investigation; Visualization. **Xiaoting Fang**: Validation; Investigation. **Shuqi Dong**: Investigation. **Jun He**: Formal analysis; Validation; Investigation. **Yilin Yang**: Validation; Investigation. **Yichen Xu**: Validation; Investigation. **Shaoxuan Fu**: Validation; Investigation. **Xujing Wang**: Investigation. **Ximing Qin**: Formal analysis; Methodology. **Qiguang Xie**: Formal analysis; Methodology. **Xiaodong Xu**: Formal analysis; Methodology. **Yuwei Zhao**: Formal analysis; Methodology. **Dan Liang**: Formal analysis; Methodology. **Peng Zhang**: Formal analysis; Methodology. **Qinfen Zhang**: Formal analysis; Methodology; Writing—review and editing. **Jinhu Guo**: Conceptualization; Data curation; Formal analysis; Funding acquisition; Writing—original draft; Writing—review and editing.

Source data underlying figure panels in this paper may have individual authorship assigned. Where available, figure panel/source data authorship is listed in the following database record: biostudies:S-SCDT-10_1038-S44318-025-00425-0.

## Disclosure and competing interests statement

The authors declare no competing interests.

# Expanded View Figures

**Figure EV1.   Cryo-EM analysis of the anKaiC.**

(**A**) Representative cryo-EM micrograph of anKaiC. (**B**) Fourier shell correlation curve of the anKaiC reconstruction. (**C**) Local resolution map of anKaiC reconstruction. (**D**) Image processing pipeline for anKaiC. Selected classes in each 3D classification are labelled by dotted line boxes. (**E**) Angular distribution of particle orientations used in the final reconstruction. (**F**) Conserved structure of partial A-loop between anKaiC (coloured in blue) and KaiC (PDB: 7S67, coloured in pink green). (**G**) The inside surface of anKaiC showing similarity to that of KaiC WT (PDB: 7S67). The surface (inside view) with identical amino acid residues in both anKaiC and KaiC are coloured in grey, the different amino acids between anKaiC and KaiC with hydrophilic residues are shown in cyan, and hydrophobic residues are shown in golden rod.

▶

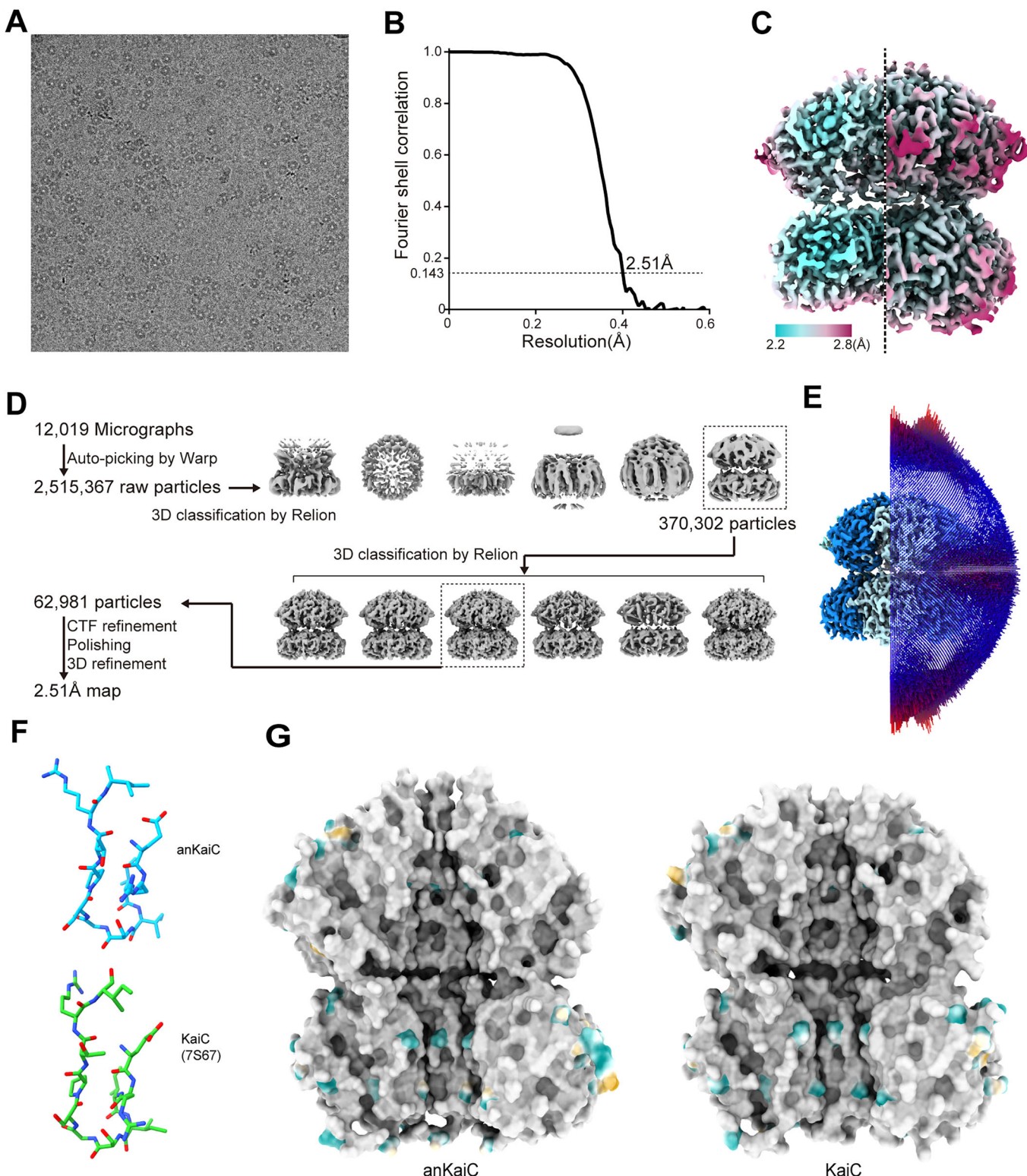

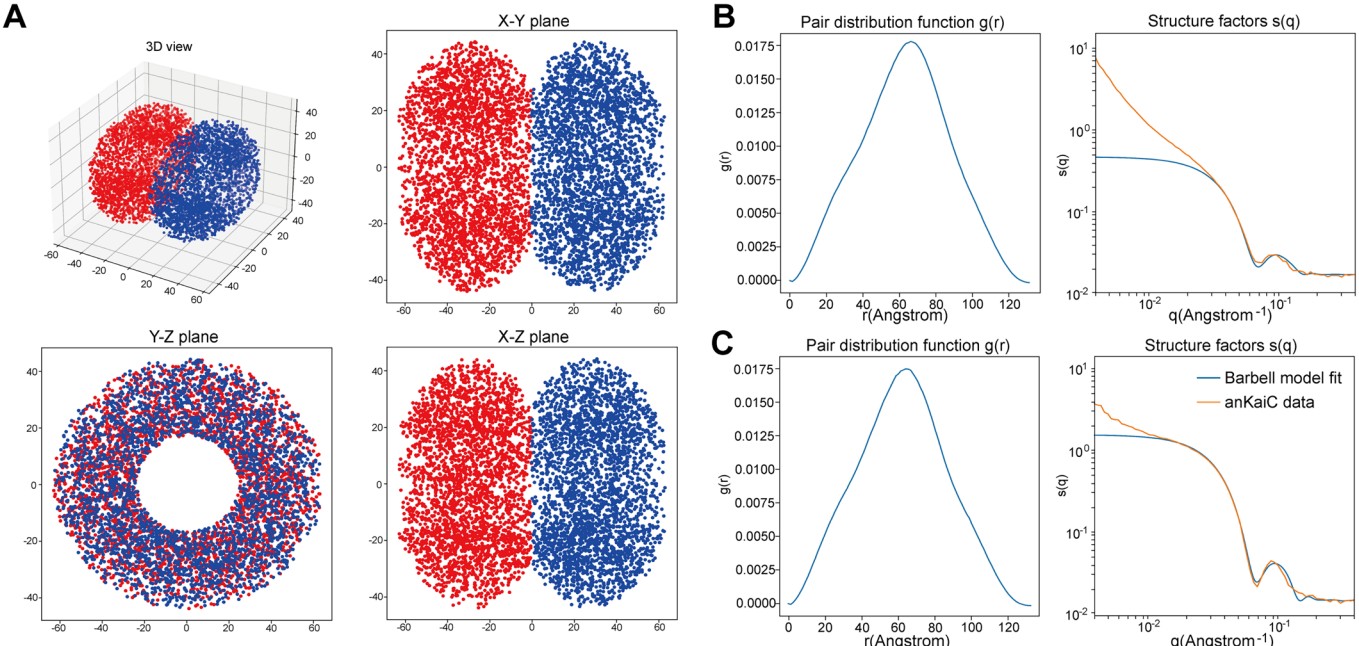

**Figure EV2. Modelling of anKaiC and KaiC SANS data.**

(A) The 3D and X, Y, Z projection of the dumbbell model of KaiC or anKaiC. (B) The pair distribution function (PDF) of the dumbbell model of KaiC and the best fit to the data with the dumbbell model. (C) The PDF of the dumbbell model of anKaiC and the best fit to the data.

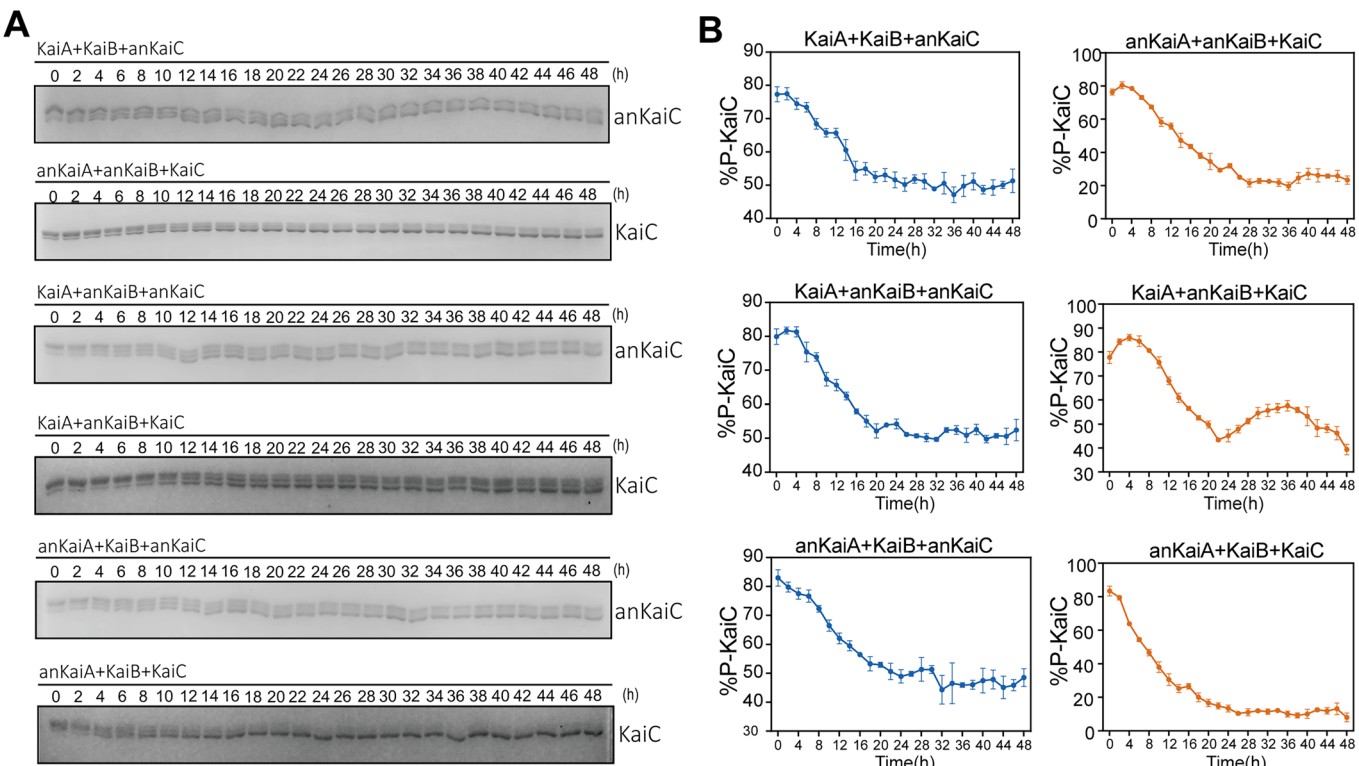

**Figure EV3.   Analysis of in vitro and in vivo rhythmicities in strains expressing different combinations of KaiABC proteins.**

(A) In vitro phosphorylation of anKaiC/KaiC proteins in different combinations. Representative results of Coomassie brilliant blue staining are shown. (B) Statistical results of bioluminescence rhythms of the indicated combinations. The percentage of hyperphosphorylated KaiC was calculated. Data are means ± SD. $n = 3$. Source data are available online for this figure.

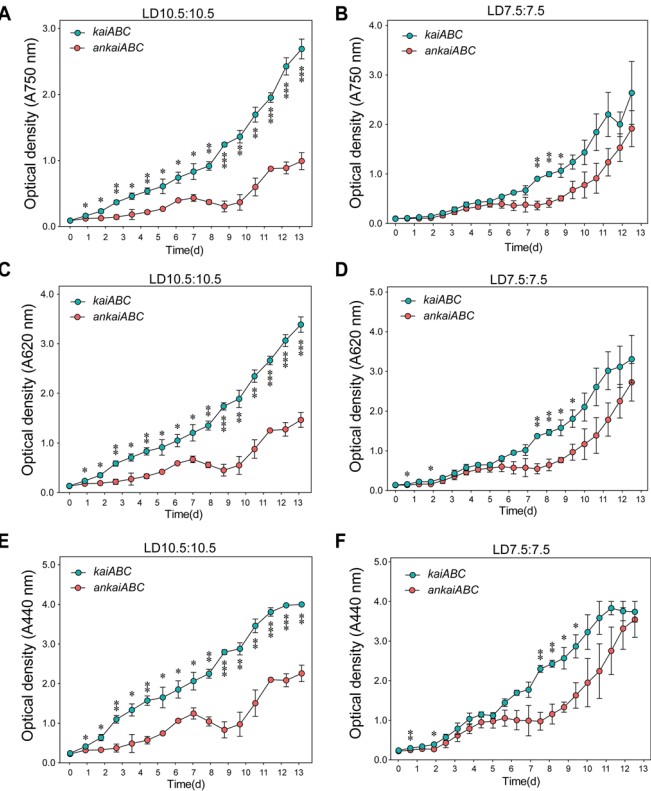

**Figure EV4. Curves of growth, phycobilin and chlorophyll a content under LD10.5:10.5 and LD7.5:7.5 conditions.**

(A, B) Growth curves of *ankaiABC* and *kaiABC* strains measured by light absorption at 750 nm under LD10.5:10.5 and LD7.5:7.5 conditions. In (A): *P* values from left to right: *P = 0.0140, *P = 0.0229, **P = 0.0065, *P = 0.0346, **P = 0.0047, *P = 0.0374, *P = 0.0147, *P = 0.0377, **P = 0.0016, ***P = 0.0004, **P = 0.0028, **P = 0.0034, ***P = 0.0001, ***P = 0.0006, ***P = 0.0010. In (B): *P* values from left to right: **P = 0.0037, **P = 0.0049, *P = 0.0187. (C, D) Phycobilin levels of *ankaiABC* and *kaiABC* strains measured by light absorption at 620 nm under LD10.5:10.5 and LD7.5:7.5 conditions. In (C): *P* values from left to right: *P = 0.0193, *P = 0.0245, **P = 0.0077, *P = 0.0367, **P = 0.0045, *P = 0.0356, *P = 0.0215, *P = 0.0391, **P = 0.0017, ***P = 0.0008, **P = 0.0062, **P = 0.0029, ***P < 0.0001, ***P = 0.0006, ***P = 0.0009. In (D): *P* values from left to right: *P = 0.0420, *P = 0.0465, **P = 0.0034, **P = 0.0074, *P = 0.0185, *P = 0.0466. (E, F) Chlorophyll a levels of *ankaiABC* and *kaiABC* strains measured by light absorption at 440 nm under LD10.5:10.5 and LD7.5:7.5 conditions. In (E): *P* values from left to right: *P = 0.0283, *P = 0.0200, **P = 0.0063, *P = 0.0367, **P = 0.0029, *P = 0.0251, *P = 0.0241, *P = 0.0361,**P = 0.0017, ***P = 0.0008, **P = 0.0050, **P = 0.0064, ***P < 0.0001, ***P = 0.0003, **P = 0.0011. In (F): *P* values from left to right: **P = 0.0056, *P = 0.0402, **P = 0.0054, **P = 0.0087, *P = 0.0146, *P = 0.0464. Data are means ± SE. *n* = 3 (independent experiments). Two-tailed unpaired Student's *t* test. Source data are available online for this figure.

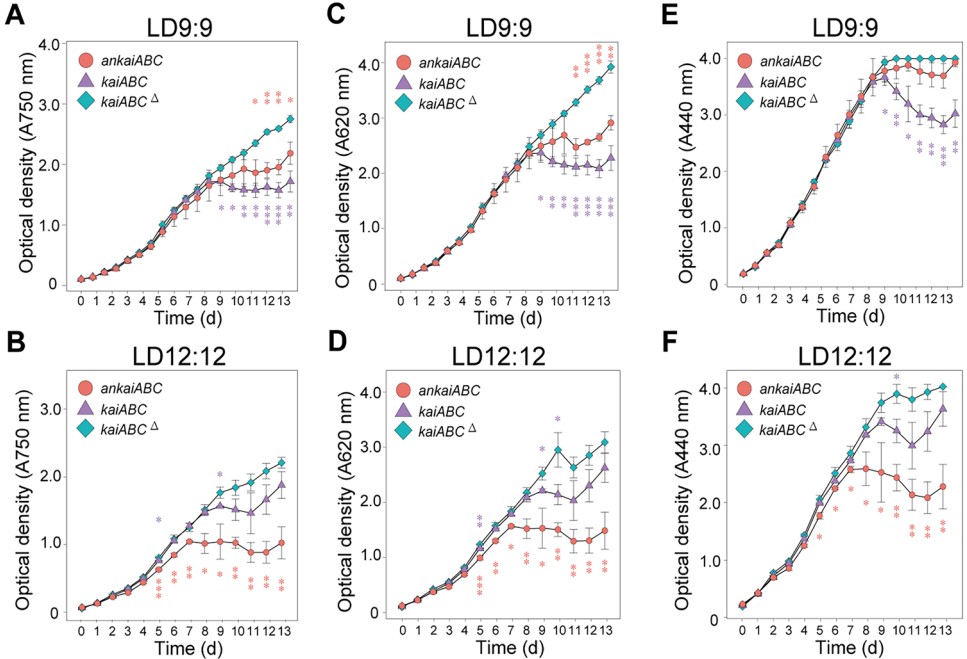

**Figure EV5.  Growth curves of cyanobacterial strains.**

(A, B) Growth curves in LD9:9 and LD12:12 measured by light absorption at 750 nm. In (A): *P* values in purple from left to right: *$P$ = 0.0202, *$P$ = 0.0184, **$P$ = 0.0015, **$P$ = 0.0010, ***$P$ = 0.0008, ***$P$ = 0.0004, **$P$ = 0.0014; *P* values in red from left to right: *$P$ = 0.0348, **$P$ = 0.0024, **$P$ = 0.0023, *$P$ = 0.0145. In (B): *P* values in purple from left to right: *$P$ = 0.0100, *$P$ = 0.0282; *P* values in red from left to right: ***$P$ = 0.0002, **$P$ = 0.0060, **$P$ = 0.0085, *$P$ = 0.0132, *$P$ = 0.0199, **$P$ = 0.0012, **$P$ = 0.0016, **$P$ = 0.0010, **$P$ = 0.0026. (C, D) Phycobilin content measured by light absorption at 620 nm. In (C): *P* values in purple from left to right: *$P$ = 0.0381, **$P$ = 0.0013, **$P$ = 0.0015, ***$P$ = 0.0007, ***$P$ = 0.0005, ***$P$ = 0.0002, ***$P$ = 0.0007; *P* values in red from left to right: **$P$ = 0.0024, ***$P$ < 0.0001, ***$P$ = 0.0002, **$P$ = 0.0010. In (D): *P* values in purple from left to right: **$P$ = 0.0023, *$P$ = 0.0216, *$P$ = 0.0338; *P* values in red from left to right: ***$P$ = 0.0009, **$P$ = 0.0034, *$P$ = 0.0111, **$P$ = 0.0095, *$P$ = 0.0212, **$P$ = 0.0037, **$P$ = 0.0027, **$P$ = 0.0016, **$P$ = 0.0042. (E, F) Content of chlorophyll a measured by light absorption at 440 nm. (A–F). In (E): *P* values in purple from left to right: *$P$ = 0.0171, **$P$ = 0.0048, *$P$ = 0.0227, **$P$ = 0.0013, **$P$ = 0.0010, ***$P$ = 0.0005, **$P$ = 0.0047. In (F): *P* values in purple from left to right: *$P$ = 0.0255; *P* values in red from left to right: *$P$ = 0.0112, *$P$ = 0.0265, *$P$ = 0.0395, *$P$ = 0.0350, *$P$ = 0.0347, **$P$ = 0.0019, **$P$ = 0.0023, **$P$ = 0.0010, **$P$ = 0.0032. Data are means ± SD, $n$ = 3 (independent experiments). Asterisks in purple denote significance between *kaiABC* and *kaiABC*$^{\triangle}$, asterisks in red denote significance between *ankaiABC* and *kaiABC*$^{\triangle}$. Two-tailed unpaired Student's *t* test. Source data are available online for this figure.

