## [Peer Review File · The EMBO Journal]

Reconstruction of the ancient cyanobacterial proto-circadian clock system KaiABC

Silin Li, Zengxuan Zhou, Yufeng Wan, Xudong Jia, Peiliang Wang, Yu Wang, Taisen Zuo, He Cheng, Xiaoting Fang, Shuqi Dong, Jun He, Yilin Yang, Yichen Xu, Shaoxuan Fu, Xujing Wang, Ximing Qin, Qiguang Xie, Xiaodong Xu, Yuwei Zhao, Dan Liang, Peng Zhang, qinfen Zhang, and Jinhu Guo

Corresponding author(s): Jinhu Guo (guojinhu@mail.sysu.edu.cn)

Review Timeline:

Submission Date:	25th May 24
Editorial Decision:	1st Jul 24
Revision Received:	29th Oct 24
Editorial Decision:	2nd Dec 24
Revision Received:	25th Jan 25
Editorial Decision:	20th Feb 25
Revision Received:	21st Feb 25
Accepted:	3rd Mar 25

Editors: Kelly M Anderson and Ioannis Papaioannou

Transaction Report:

Dear Dr. Guo,

Thank you for submitting your manuscript for consideration by the EMBO Journal. Thank you also for providing a preliminary plan to address the referee concerns, which I believe sounds reasonable, therefore I would like to invite you to submit a revised version of the manuscript, addressing the comments of all three reviewers. I should add that it is EMBO Journal policy to allow only a single round of revision, and acceptance of your manuscript will therefore depend on the completeness of your responses in this revised version.

Thank you for the opportunity to consider your work for publication. I look forward to your revision.

Yours sincerely,

Kelly M Anderson, PhD
Editor, The EMBO Journal
k.anderson@embojournal.org

We realize that it is difficult to revise to a specific deadline. In the interest of protecting the conceptual advance provided by the work, we recommend a revision within 3 months (29th Sep 2024). Please discuss the revision progress ahead of this time with the editor if you require more time to complete the revisions. Use the link below to submit your revision:

Referee #1:

This study explores how ancient organisms adapted to the Earth's shorter day-night cycle through the lens of the KaiABC circadian system. By resurrecting the ancestral KaiABC genes (anKaiABC) from 0.95 billion years ago, the researchers discovered that while this ancient system lacked self-sustained oscillation, it could still be entrained by an 18-hour light/dark cycle. This suggests that early organisms may not have possessed an intrinsic circadian clock based on KaiABC, but rather relied on an adaptable system to respond to the environmental cues of the faster rotation. These findings challenge our assumptions regarding the evolution of KaiABC-based circadian rhythms and shed light on the remarkable adaptability of ancient life.

Several key concerns need to be addressed to fully support the study's conclusions regarding the lack of an endogenous circadian clock in the ancient organism and its adaptation to the 18-hour day-night cycle.

1. The study suggests the anKaiABC system lacked an intrinsic clock yet could adapt to an 18-hour light/dark cycle. However, the mechanism for this adaptation remains unclear. Without a self-sustaining oscillation, it's difficult to understand how the system could adjust its activity to match the 18-hour cycle. Further investigation into potential mechanisms, such as alternative light-sensitive pathways influencing the anKaiABC system, and explanations are necessary to solidify this claim.
2. The study focuses on the anKaiABC system, but the conclusion regarding the lack of an intrinsic clock in early organisms is broader. It's crucial to clarify if this finding applies solely to KaiABC or casts doubt on the existence of intrinsic clocks in other circadian systems altogether.
3. The study identifies "subtle" structural differences between anKaiC and its modern counterpart through cryo-EM analysis. However, the link between these structural variations and the observed functional differences (lower activity and lack of endogenous oscillation) remains unclear. Can the authors please specify what "subtle" structural differences are? Are they all important or which are drivers vs passengers? Elucidating the specific molecular mechanisms by which these structural changes might impact protein function would be essential to solidify the connection between structure and observed phenotypes in the strain competition experiments. Otherwise, this reviewer is left with the impression that cryo-EM structures are not even as useful as classic genetic approaches/mutagenesis experiments in answering functional questions.
Minor concerns: writing and organization of logical flow can be much improved to enhance readability and highlight the potential significance of the findings for a broader audience.

Referee #2:

The manuscript from Li et al. explores a fascinating concept: since the general belief about circadian clocks is that their periods should be tuned to closely match the rotational period of the Earth, ancestral reconstruction of clock genes may provide a window into the function of these systems when the rotational speed of the Earth was known to be much faster. Despite the novelty and intrinsic interest of this approach, I have major concerns about the interpretation of the work and whether the main conclusions are supported.

1. The process of ancestral reconstruction is a statistical procedure and multiple candidate sequences at multiple nodes of the phylogenetic tree can be found. This process is presumably more complex when trying to reconstruct three genes (kaiABC). Since the study only presents results on a single reconstructed sequence, it is unclear at this point how robust the biochemical and competition experiment results are, or whether these are idiosyncratic to a specific choice of reconstruction.
2. The slow phosphorylation and ATPase rate of the anKaiC are somewhat paradoxical. The general concept in the existing literature is that the catalytic rate of KaiC tends to scale with the frequency of the cycle. Since the conclusion is that this anKaiC is adapted to an 18 hour cycle, it is unexpected that catalytic rates are slower. Of course this finding could be correct, but it suggests the alternative interpretation that the anKaiC has a partial loss of function.
3. Related to this, kaiABC-null wins in the 9:9 competition compared to both WT and the reconstructed strain. This suggests that

loss of KaiC function may be beneficial in short LD cycles. More evidence is needed to determine whether the ancestral KaiABC system is truly "adapted" to 9:9 vs the WT being defective. For example, testing other LD periods to show a peak specifically around the 18 hour day.

4. kaiABC-null wins in the experimental conditions regardless of LD cycle. This is counter to previous work from the Johnson lab and makes the interpretation difficult. Considered on its own terms, the conclusion from the data shown here should be that the kaiABC system is deleterious in LD cycles. This seems counter to the main conclusion in the title & abstract

Referee #3:

The authors investigated here the evolution of the circadian clock in cyanobacterial species. They reconstructed a potential precursor circadian clock from a potential ancestor living 0.95 Ga ago based on a phylogenetic tree and characterized its structure and functioning in the context of a modern circadian clock. The main finding was that this ancient circadian clock, although very much related to the modern clock, was not able to confer self-sustained circadian rhythms. This particular trait arose later in the evolution of the cyanobacteria. Surprisingly, though, the ancient circadian clock mechanism provided better fitness to persist in a 9h:9h light-dark cycle, which was present 0.95 Ga ago. Altogether, the ancient circadian clock, although immature compared to the modern clock, provided some evolutionary advantage for these early cyanobacteria. The study is well done, original, and the conclusions overall valid. The manuscript needs some minor revision, though.

A) Figure 1A: why is this phylogenetic tree based on different organisms than Figs. 1B and C and gives a similar time estimate? Did the authors verify with some other, unrelated genes? In the materials and methods, it is not clear, which method was used to create the phylogenetic trees (compare beginning and end of the paragraph)? Please provide a table with the name of the sequence files used for the construction and the parameters used within the programs. Afterall, the construction of the phylogenetic tree and how to obtain the ancient KaiABC protein sequences should be reproducible.

B) In general, the statistical tests applied seem not to be appropriate. A T-test can only compare two data sets. If using more comparisons, then it has to be corrected for multiple testing. In most figures, a two-way ANOVA should be used, if comparing two data sets over multiple time points, or some kind of regression model.

C) Figure 2F should probably read KaiC (instead of 7S67)?

D) Line 446: ultrathin sections (how many um)?

Point-to-point replies to reviewers' comments

Referee 1:

This study explores how ancient organisms adapted to the Earth's shorter day-night cycle through the lens of the KaiABC circadian system. By resurrecting the ancestral KaiABC genes (*anKaiABC*) from 0.95 billion years ago, the researchers discovered that while this ancient system lacked self-sustained oscillation, it could still be entrained by an 18-hour light/dark cycle. This suggests that early organisms may not have possessed an intrinsic circadian clock based on KaiABC, but rather relied on an adaptable system to respond to the environmental cues of the faster rotation. These findings challenge our assumptions regarding the evolution of KaiABC-based circadian rhythms and shed light on the remarkable adaptability of ancient life.

Several key concerns need to be addressed to fully support the study's conclusions regarding the lack of an endogenous circadian clock in the ancient organism and its adaptation to the 18-hour day-night cycle.

1. The study suggests the *anKaiABC* system lacked an intrinsic clock yet could adapt to an 18-hour light/dark cycle. However, the mechanism for this adaptation remains unclear. Without a self-sustaining oscillation, it's difficult to understand how the system could adjust its activity to match the 18-hour cycle. Further investigation into potential mechanisms, such as alternative light-sensitive pathways influencing the *anKaiABC* system, and explanations are necessary to solidify this claim.

Re: Indeed, there are many microorganisms contain no endogenous self-sustaining circadian clock system in the wild or in our gut, e.g., *E. coli* and yeast. However, they still show oscillations with the 24-hour cycling cues though whether the 24-h period is preferable remains little known, which definitely deserves further investigation. Some of the cyanobacteria harbor only partial components of the circadian system, and they have the hourglass-type oscillator [Hochberg et al *Annu Rev Biophys.* 2017;46:247-69; Ma et al *PLoS Genet.* 2016;12:e1005922; Pitsawong et al *Nature.* 2023;616:183-9], but perhaps they may have some unknown mechanisms to adapt to the 24-hour cycles. There are some facts demonstrating that some gut microorganisms can be entrained by factors secreted from hosts [Paulose et al *Sci Rep.* 2016;11(1):e0146643; Thaiss Cet al 2014 *Cell*;159(3):514-29], which implies that the 24-hour cycles of the host physiology may be also preference of these microorganisms. Of course, this issue definitely needs validation in the future as it is very important.

In terms of the light-sensitive pathway, it may not determine oscillation and period of the circadian system. As evidenced in modern circadian systems, basically it is the transcription-translational negative feedback loop (TTFL) and/or translational oscillation mechanism (PTO) that drive the circadian rhythms and the input pathway functions to entrain the circadian rhythms [Hurley et al. *Trends Biochem Sci.* 2016;41(10):834-846]. In this study, we also checked the rhythmicity of the *anKaiABC* strain which was generated through transforming the *anKaiABC* genes into the *KaiABC^Δ* strain. In this scenario, the light-sensitive pathways are the same in the three strains *KaiABC*, *anKaiABC*, and *KaiABC^Δ*. However, these three strains displayed differentially under different light-dark conditions suggesting that the core

circadian clock genes KaiABC/anKaiABC predominantly determine – if not entirely – the circadian rhythms and adaptation to the cycling environmental cues.

2. The study focuses on the anKaiABC system, but the conclusion regarding the lack of an intrinsic clock in early organisms is broader. It's crucial to clarify if this finding applies solely to KaiABC or casts doubt on the existence of intrinsic clocks in other circadian systems altogether.

Re: We concluded that way since there were only prokaryotic organisms and simple eukaryotic algae ~0.95 billion years ago. Of course, to be more prudential, here we agree with you and we have revised this conclusion and specify it only on cyanobacteria (last paragraph in discussion).

3. The study identifies "subtle" structural differences between anKaiC and its modern counterpart through cryo-EM analysis. However, the link between these structural variations and the observed functional differences (lower activity and lack of endogenous oscillation) remains unclear. Can the authors please specify what "subtle" structural differences are? Are they all important or which are drivers vs passengers? Elucidating the specific molecular mechanisms by which these structural changes might impact protein function would be essential to solidify the connection between structure and observed phenotypes in the strain competition experiments. Otherwise, this reviewer is left with the impression that cryo-EM structures are not even as useful as classic genetic approaches/mutagenesis experiments in answering functional questions.

Re: We agree that the structure of anKaiC is not dramatically different from KaiC. However, even the subtle difference may lead to big change. For instance, the subtle difference and angle in the position of the key amino acid residues in the ATP binding region determine the efficiency of the ATPase activity of KaiC [Abe et al. *Science*. 2015;349(6245):312-316]. In addition, now we are carrying out very small angle neutron scattering (VSANS) [Sato et al. *Sci Rep*. 2016;6:35567; Zuo et al. *J Appl Crystallogr*. 2024;57(Pt 2):380-391] experiments to compare the structural differences between KaiABC and anKaiABC proteins, and the results also revealed structural differences between kaiC/anKaiC proteins and KaiAC/KaiAC mixtures (Fig. 2H-J).

Moreover, during revision we carried out experiments to generate mutants in the B-loop (KaiC-V118A) and ATP-binding (S239T, T240N, and S239T/T240N) regions in KaiC to assess their impacts on the differential activity of KaiC. The results showed that the double mutations of KaiC-S229T/T240N compromised its ATPase activity and probably kinase and phosphatase activities as well. However, the KaiC-V118A mutation showed no significant influence on its function in promoting KaiC desphorylation which may be explained by that the highly flexible structure of B-loop is not affected by this mutation (Fig. 4A-E). As the structure of anKaiC A-loop is very similar to that of anKaiC (Fig. EV2F), therefore we did not work on this region. The results from these experiments may help us understand the underlying mechanisms.

Minor concerns: writing and organization of logical flow can be much improved to enhance readability and highlight the potential significance of the findings for a broader audience.

Re: We have had the language improved by the language service AJE company, and we also have

reorganized some of the contents to enhance readability and highlight the potential significance by following your suggestions.

Referee 2:

The manuscript from Li et al. explores a fascinating concept: since the general belief about circadian clocks is that their periods should be tuned to closely match the rotational period of the Earth, ancestral reconstruction of clock genes may provide a window into the function of these systems when the rotational speed of the Earth was known to be much faster. Despite the novelty and intrinsic interest of this approach, I have major concerns about the interpretation of the work and whether the main conclusions are supported.

1. The process of ancestral reconstruction is a statistical procedure and multiple candidate sequences at multiple nodes of the phylogenetic tree can be found. This process is presumably more complex when trying to reconstruct three genes (*kaiABC*). Since the study only presents results on a single reconstructed sequence, it is unclear at this point how robust the biochemical and competition experiment results are, or whether these are idiosyncratic to a specific choice of reconstruction.

Re: Yes, there are more than one node. The reason that we picked the common node of ~0.95 billion years ago is because we wanted to predict and reconstruct the ancient genes as early as possible, and 0.95 billion years is the only common node for KaiABC proteins (Fig. 1A-C). And we have provided supplemental material including all the ancient sequences at different nodes in the revised version (source data for Fig. 1A-C).

2. The slow phosphorylation and ATPase rate of the *anKaiC* are somewhat paradoxical. The general concept in the existing literature is that the catalytic rate of KaiC tends to scale with the frequency of the cycle. Since the conclusion is that this *anKaiC* is adapted to an 18-hour cycle, it is unexpected that catalytic rates are slower. Of course, this finding could be correct, but it suggests the alternative interpretation that the *anKaiC* has a partial loss of function.

Re: You are right, the cyanobacterial ATPase activity of KaiC defines the circadian period [Terauchi et al. Proc Natl Acad Sci U S A. 2007;104(41):16377-81]. However, *anKaiABC* system shows no self-sustaining in vitro circadian rhythms of *anKaiC* phosphorylation status and the *anKaiABC* strain is also arrhythmic. The correlation between ATPase rate and circadian period may be applied to KaiABC systems or strains with endogenous circadian rhythmicity but not those systems or strains containing no endogenous circadian rhythms, for example, the *anKaiABC* system or *anKaiABC* strain. As further shown in the new data in Figure 7, the *anKaiABC* strain defects the *KaiABC* strain in all tested short cycles, not specifically under LD9:9. In addition, the results of ATPase assays have been changed to be demonstrated as “produced ADP (KaiC⁻¹)” (Fig. 3E and Fig. 4F).

3. Related to this, *kaiABC*-null wins in the 9:9 competition compared to both WT and the reconstructed strain. This suggests that loss of KaiC function may be beneficial in short LD cycles. More evidence is needed to determine whether the ancestral KaiABC system is truly "adapted" to 9:9 vs the WT being

defective. For example, testing other LD periods to show a peak specifically around the 18-hour day.

Re: To assess the effects of more LD cycles on the adaptation of ancient and contemporary cyanobacteria, we have carried out experiments to monitor the growth and contents of photosynthesis-related pigments of *KaiABC* and *anKaiABC* strains under LD10.5:10.5 and LD7.5:7.5 conditions. The results showed greater levels of growth, phycobilin, and chlorophyll a under LD12:12, LD10.5:10.5 and LD7.5:7.5 conditions. In contrast, the ancestral cyanobacteria showed higher levels of growth, phycobilin, and chlorophyll a under LD9:9 after ~10-day growth (Fig. 6, Fig. EV6). Together, these data suggest that the ancestral cyanobacterial clock can be a short day-length although it may not be specific to the day length of 18 h.

4. *kaiABC*-null wins in the experimental conditions regardless of LD cycle. This is counter to previous work from the Johnson lab and makes the interpretation difficult. Considered on its own terms, the conclusion from the data shown here should be that the *kaiABC* system is deleterious in LD cycles. This seems counter to the main conclusion in the title & abstract.

Re: I assume you mean the results in supplementary Figure S4. The experiments described in this figure is a little different from Johnson's work. These curves represent the growth rate instead of competition test, which only indicate that the *KaiABC*-null strain grew and reproduced faster after one week in the flask which the nutrition may have been exhausted. The purpose that we show supplementary Figure S4 is to demonstrate that the *anKaiABC* strain does not equal to the *KaiABC*-null strain, suggesting that the *anKaiABC* system already possesses some functions through its circadian function had not been mature yet. Therefore, these data only suggest that the *KaiABC*-null strain grows well in the exhausted condition solely, but it does not necessarily mean it is the most competitive when they are mixed. And in the competition test we did not include the *KaiABC*-null strain.

Referee 3:

The authors investigated here the evolution of the circadian clock in cyanobacterial species. They reconstructed a potential precursor circadian clock from a potential ancestor living 0.95 Ga ago based on a phylogenetic tree and characterized its structure and functioning in the context of a modern circadian clock. The main finding was that this ancient circadian clock, although very much related to the modern clock, was not able to confer self-sustained circadian rhythms. This particular trait arose later in the evolution of the cyanobacteria. Surprisingly, though, the ancient circadian clock mechanism provided better fitness to persist in a 9h:9h light-dark cycle, which was present 0.95 Ga ago. Altogether, the ancient circadian clock, although immature compared to the modern clock, provided some evolutionary advantage for these early cyanobacteria.

The study is well done, original, and the conclusions overall valid. The manuscript needs some minor revision, though.

A) Figure 1A: why is this phylogenetic tree based on different organisms than Figs. 1B and C and gives a similar time estimate? Did the authors verify with some other, unrelated genes? In the materials and methods, it is not clear, which method was used to create the phylogenetic trees (compare beginning and

end of the paragraph)? Please provide a table with the name of the sequence files used for the construction and the parameters used within the programs. After all, the construction of the phylogenetic tree and how to obtain the ancient KaiABC protein sequences should be reproducible.

Re: In the reconstruction of ancient cyanobacterial clock proteins through phylogenetic tree, we tried to include as many modern proteins as input as possible. However, some of the modern cyanobacteria contain no KaiA but most, if not all, contain KaiB and KaiC [Hochberg et al *Annu Rev Biophys.* 2017;46:247-69; Ma et al *PLoS Genet.* 2016;12:e1005922; Pitsawong et al *Nature.* 2023;616:183-9]. Furthermore, in the evolution of Cyanobacterial clock system, KaiA might have appeared in cyanobacteria genomes 1.4 - 1.6 Ga ago, much later than KaiB and KaiC [Baca et al. *J Mol Evol.* 2010;70:453-65]. These are explanations why the quantity of organisms in Fig. 1A is less than those in Fig. 1B and Fig. 1C. We have provided the information of the input sequences and the ancient sequence at different time nodes in the revised version and the accession date to the database (source data for Fig. 1A-C).

B) In general, the statistical tests applied seem not to be appropriate. A T-test can only compare two data sets. If using more comparisons, then it has to be corrected for multiple testing. In most figures, a two-way ANOVA should be used, if comparing two data sets over multiple time points, or some kind of regression model.

Re: We have checked through this issue, and since in this manuscript we only compared the two data sets or data sets at different time points. We did not compare the overall results over multiple time points. Therefore, only *T*-test was used in this study.

C) Figure 2F should probably read KaiC (instead of 7S67)?

Re: We have fixed it to be “KaiC” in the revised version (Fig. 2F).

D) Line 446: ultrathin sections (how many μm)?

Re: We have added the related information. The thickness of ultrathin sections was $\sim 50 - 100 \text{ nm}$ (the last sentence in section “Transmission electron microscopy (TEM) sample preparation and results analysis”, in the methods section). In addition, the scale bar in Fig 6. Legend denotes 100 nm (Fig. 6 legend).

Dear Dr. Guo,

Thank you again for submitting your revised manuscript (EMBOJ-2024-117992R) to The EMBO Journal for our consideration. As I have already informed you, it has now been seen by the three referees who previously assessed the original version of your manuscript, and we have received their comments, which are included below.

Referees #1 and #3 are satisfied with the revision and mention that their previous concerns have been sufficiently addressed. On the other hand, referee #2 points out that the main conclusions of the manuscript are not fully supported by the presented data, as other alternative interpretations of the results cannot be excluded.

I would like to thank you for your responses to the comments of referee #2 that you shared with me. You provided further explanations on the concerns raised by the referee, and you expressed your willingness to strengthen your manuscript further by performing additional experimental work and/or toning down the conclusions of your manuscript as necessary.

In light of the input we received from the referees and your willingness to address the remaining concerns of referee #2 as well, I would like to invite you to perform the additional experiments you suggested and submit another revised version of the manuscript by the end of January 2025, along with a detailed point-by-point response to the referees' comments. I should note that acceptance of your manuscript will depend on the completeness of your responses in this final version of the manuscript.

There are also some formatting/editorial changes that we would kindly request you to make in your final revision before resubmission:

- Please remove "Title page" from the first page of the manuscript.
- Please remove the Figures from the main manuscript file and only upload them to our manuscript handling system as individual, high-resolution Figure files.
- The Expanded View (EV) Figures (typically up to 5) should also be uploaded as individual, high-resolution Figure files.
- All Figure and EV Figure legends should remain in the manuscript file.
- Please include in your Data availability statement the specific links (URLs) to your deposited datasets.
- The author contributions statement should be removed from the manuscript file. Instead, we use CRediT to specify the contributions of each author in the journal submission system. Please feel free to use the free text box to provide more detailed descriptions during submission. See also our guide to authors for more information:
<https://www.embopress.org/page/journal/14602075/authorguide#authorshipguidelines>.
- We noticed that callouts for Figure EV5 are missing; please make sure that all Figures and their panels are called out in the main manuscript file.
- Please upload a complete author checklist, which you can download from our author guidelines (<https://www.embopress.org/page/journal/14602075/authorguide>). Please note that the checklist will also be part of the Review Process File.
- The Appendix should be uploaded as a single PDF file containing Figures and Tables with the nomenclature: "Appendix Figure S#" and "Appendix Table S#", respectively. Please update the nomenclature throughout the Appendix and all their callouts in the main manuscript file. In addition, please add the manuscripts' title below "Appendix" on its first page, and include a brief Table of Contents with page numbers on the same page of the Appendix. Regarding your previous "Table EV2", please move this to the Reagents and Tools Table (see next point for more information).
- All materials and methods need to be described in the manuscript using our structured methods format, which is now required for all research articles. According to this format, the Methods section includes a single "Reagents and Tools Table" -listing key reagents, experimental models, software and relevant equipment including their sources and relevant identifiers- followed by a "Methods and Protocols" section describing the methods. Please download and fill our Reagents and Tools Table template (.docx), which you can find in our author guide:
<https://www.embopress.org/page/journal/14602075/authorguide#structuredmethods>. When submitting your revised manuscript, please do not include the Reagents and Tools Table in the Methods section of the manuscript but instead upload it as a separate file choosing the file type "Reagent Table".
- The source data (including gel images etc.) should be organized in and uploaded as a single ZIP folder per Figure (e.g. all source data files for the panels of Figure 1 should be saved in a single ZIP folder named "SD Figure 1.zip"). For EV and/or

Appendix Figures, please ZIP together all their source data in a single "SD Expanded View.zip" folder. Before resubmission of your manuscript, please complete the SD checklist that our Source Data curator has compiled, and upload it along with your source data, as a "Related Manuscript File". If you have deposited some of the requested Source Data to other repositories (such as BioStudies) instead of uploading them to our manuscript handling system, please explain so in your SD checklist.

- Please note that EMBO press papers are accompanied online by:

A) a short (2 sentences) summary of the findings and their significance, and

B) 2-5 short bullet points highlighting the key results, and

C) a synopsis image in .jpg or .png format that is exactly 550 pixels wide and 300-600 pixels high (the height is variable). Please note that the text needs to be legible at the final size.

Please upload this information along with your revised manuscript (the text for A and B should be provided in a separate Word file).

- During our standard Figure checks, our Data Integrity analyst detected blot re-use in your Figure 4E. In particular, the cropped blot bands provided for "anKaiC" are identical to the ones shown for "KaiC-S229T-T240N". If this is intentional, please explain it in the Figure legend. If it is a mistake, please correct it in your revised manuscript/figure. Please also provide the uncropped blots in your Source Data.

- During our routine pre-acceptance checks, our data editors have raised the following queries regarding figures, data, and legends. Please completely address all requests in the final version of your manuscript:

1. Please indicate what */ **/ ***/ **** represents; if this represents p value(s), please indicate the statistical test used and the exact p value in the legends of Figures 3D, E; 4B-F; 5D; 6A, B, C, D, E, F, K, L; 7C.

2. Please note that information related to "n" is missing in the legend of Figure 7C.

3. Although "n" is provided, please describe the nature of entity for "n" in the legends of Figures 3A-E; 4B-F; 5C, D; 6A-F.

- Please also change the manuscript section order as follows: Title page - Abstract & Keywords - Introduction - Results - Discussion - Methods - Data Availability - Acknowledgements - Disclosure and Competing Interests Statement - References - Figure Legends - main Tables (not EV Tables or Appendix Tables) - Expanded View Figure Legends.

Please also note that as part of the EMBO publications' Transparent Editorial Process, The EMBO Journal publishes online a Peer Review File along with each accepted manuscript. This File will be published in conjunction with your paper and will include the referee reports, your point-by-point response and all pertinent correspondence relating to the manuscript. You can opt out of this by letting the editorial office know (contact@embojournal.org). If you do opt out, the Peer Review File link will point to the following statement: "No Peer Review File is available with this article, as the authors have chosen not to make the review process public in this case."

We look forward to seeing a final version of your manuscript as soon as possible. Please let us know if you have any questions and use this link to submit your revision when you are ready: <https://emboj.msubmit.net/cgi-bin/main.plex>.

Best regards,

Ioannis

Referee #1:

The authors provided extensive revision and satisfactorily addressed my previous concerns.

Referee #2:

The manuscript by Li et al. presents interesting results on the properties of Kai proteins derived from an ancestral reconstruction process. However, I cannot endorse publication of this paper in its current form because the headline claim that the ancient kai system is adapted to a short day period is not strongly supported by the data and other interpretations are not only possible but likely.

If we consider the data independent of the faster rotation speed of ancient Earth, we see:

- * ancestrally reconstructed Kai proteins do not support free-running oscillation
- * the ATPase rate of ancestral KaiC is slow, suggesting that the response time of the system is long
- * the growth advantage that WT (extant kaiABC genes) has in 12:12 LD cycles is no longer present in 9:9 cycles. But this is expected solely based on the idea that the WT circadian oscillator is beneficial only when matched to the environmental frequency and does not itself indicate that there is a matching condition for the ancestral reconstructed genes. (indeed the kaiABC-null strain outperforms in the conditions shown here)

I would only consider the title of the paper to be justified by experimental evidence if:

- * the ancestral genes create a ~18 h oscillation (in vivo or in vitro)

or

- * the strain with the ancestral genes shows fitness vs. environmental frequency that is sharply peaked around an 18 h day

the data in this manuscript, while interesting, may well indicate that the ancient kai genes had a role quite different from their modern function in circadian rhythms.

A secondary but related concern is that it is unclear whether the properties seen here are idiosyncratic to the particular ancestral sequence selected from maximum likelihood estimation or whether these conclusions are robust when considering other nodes on the tree.

Referee #3:

The authors have addressed all my concerns.

Point-by-point replies to Editor's comments

There are also some formatting/editorial changes that we would kindly request you to make in your final revision before resubmission:

1.- Please remove "Title page" from the first page of the manuscript.

Re: We have removed it.

2. - Please remove the Figures from the main manuscript file and only upload them to our manuscript handling system as individual, high-resolution Figure files.

Re: We have removed the figures from the main manuscript and uploaded them separately.

3. - The Expanded View (EV) Figures (typically up to 5) should also be uploaded as individual, high-resolution Figure files.

Re :We have uploaded the EV figures separately by following your guides. And we moved the previous EV figures EV1 and EV3 to Appendix as appendix Fig. S1 and S2, respectively. In addition, the citations of the EV figures and these two new appendix figures in the manuscript have also been updated.

4. - All Figure and EV Figure legends should remain in the manuscript file.

Re: We have checked and confirmed this issue.

5. - Please include in your Data availability statement the specific links (URLs) to your deposited datasets.

Re: It has been done.

6. - The author contributions statement should be removed from the manuscript file. Instead, we use CRediT to specify the contributions of each author in the journal submission system. Please feel free to use the free text box to provide more detailed descriptions during submission. See also our guide to authors for more information: <https://www.embopress.org/page/journal/14602075/authorguide#authorshipguidelines>.

Re: We have removed this part from the manuscript and used CRediT to specify it in the submission system.

7. - We noticed that callouts for Figure EV5 are missing; please make sure that all Figures and their panels are called out in the main manuscript file.

Re: We have added the callout for Figure EV5 in the sentence "In contrast, all of the other combinations presented no detectable rhythms, although anKaiB promoted KaiC phosphorylation at 30 - 40 h together with KaiA (Fig. 5A,B; Fig. EV3A,B)".Since only five

EV figures are allowed, we have removed previous Fig. EV1 and EV4, therefore the previous EV5 is now EV3 in the updated manuscripts.

8. - Please upload a complete author checklist, which you can download from our author guidelines (<https://www.embopress.org/page/journal/14602075/authorguide>). Please note that the checklist will also be part of the Review Process File.

Re: We have completed and uploaded the author checklist.

9. - The Appendix should be uploaded as a single PDF file containing Figures and Tables with the nomenclature: "Appendix Figure S#" and "Appendix Table S#", respectively. Please update the nomenclature throughout the Appendix and all their callouts in the main manuscript file. In addition, please add the manuscripts' title below "Appendix" on its first page, and include a brief Table of Contents with page numbers on the same page of the Appendix. Regarding your previous "Table EV2", please move this to the Reagents and Tools Table (see next point for more information).

Re: We have added an appendix file, with the appendix figures S1 and S2 depicting the SDS-polyacrylamide gel electrophoresis results. Appendix Figures S1 and S2 are previous Fig. EV1 and EV4, which were moved here to make sure that the number of Expanded View (EV) Figures is less than five (# 3).

10. - All materials and methods need to be described in the manuscript using our structured methods format, which is now required for all research articles. According to this format, the Methods section includes a single "Reagents and Tools Table" -listing key reagents, experimental models, software and relevant equipment including their sources and relevant identifiers- followed by a "Methods and Protocols" section describing the methods. Please download and fill our Reagents and Tools Table template (.docx), which you can find in our author guide: <https://www.embopress.org/page/journal/14602075/authorguide#structuredmethods>.

When submitting your revised manuscript, please do not include the Reagents and Tools Table in the Methods section of the manuscript but instead upload it as a separate file choosing the file type "Reagent Table".

Re: We have added the reagents and tools table and uploaded it.

11. - The source data (including gel images etc.) should be organized in and uploaded as a single ZIP folder per Figure (e.g. all source data files for the panels of Figure 1 should be saved in a single ZIP folder named "SD Figure 1.zip"). For EV and/or Appendix Figures, please ZIP together all their source data in a single "SD Expanded View.zip" folder. Before resubmission of your manuscript, please complete the SD checklist that our Source Data curator has compiled, and upload it along with your source data, as a

"Related Manuscript File". If you have deposited some of the requested Source Data to other repositories (such as BioStudies) instead of uploading them to our manuscript handling system, please explain so in your SD checklist.

Re: We have reorganized and uploaded the data of figures, EV figures and related files by following the guidance.

12. - Please note that EMBO press papers are accompanied online by:

A) a short (2 sentences) summary of the findings and their significance, and

B) 2-5 short bullet points highlighting the key results, and

C) a synopsis image in .jpg or .png format that is exactly 550 pixels wide and 300-600 pixels high (the height is variable). Please note that the text needs to be legible at the final size.

13. - Please upload this information along with your revised manuscript (the text for A and B should be provided in a separate Word file).

Re: In terms of A) and B), separate files of short summary and bullet points have been uploaded as instructed. And in terms of C), the new synopsis image with modification has been uploaded.

14. - During our standard Figure checks, our Data Integrity analyst detected blot re-use in your Figure 4E. In particular, the cropped blot bands provided for "anKaiC" are identical to the ones shown for "KaiC-S229T-T240N". If this is intentional, please explain it in the Figure legend. If it is a mistake, please correct it in your revised manuscript/figure. Please also provide the uncropped blots in your Source Data.

Re: It is a mistake, and we have updated the correct image and the uncropped blots have been included in the related source data file.

15. - During our routine pre-acceptance checks, our data editors have raised the following queries regarding figures, data, and legends. Please completely address all requests in the final version of your manuscript:

(1) Please indicate what */ **/ ***/ **** represents; if this represents p value(s), please indicate the statistical test used and the exact p value in the legends of Figures 3D, E; 4B-F; 5D; 6A, B, C, D, E, F, K, L; 7C.

Re: We have done the modifications by following your suggestions.

(2) Please note that information related to "n" is missing in the legend of Figure 7C.

Re: The missing information has been added as "n=3, ~100 colonies".

(3) Although "n" is provided, please describe the nature of entity for "n" in the legends of

Figures 3A-E; 4B-F; 5C, D; 6A-F.

Re: It has been done in these figure legends and some legends of other figures (Fig. EV4 and Fig. EV5).

16. - Please also change the manuscript section order as follows: Title page - Abstract & Keywords - Introduction - Results - Discussion - Methods - Data Availability - Acknowledgements - Disclosure and Competing Interests Statement - References - Figure Legends - main Tables (not EV Tables or Appendix Tables) - Expanded View Figure Legends.

Re: We have checked and reorganized the manuscript section according to this order.

17. - Please also note that as part of the EMBO publications' Transparent Editorial Process, The EMBO Journal publishes online a Peer Review File along with each accepted manuscript. This File will be published in conjunction with your paper and will include the referee reports, your point-by-point response and all pertinent correspondence relating to the manuscript. You can opt out of this by letting the editorial office know (contact@embojournal.org). If you do opt out, the Peer Review File link will point to the following statement: "No Peer Review File is available with this article, as the authors have chosen not to make the review process public in this case."

Re: It will be fine to publish a peer review file online along with each accepted manuscript.

18. – Additional revisions:

By referring to more literatures, we revised the strain names of “*KaiABC*” and “*anKaiABC*” to “*kaiABC*” and “*ankaiABC*”, i.e., the initials are in lowercase.

Point-by-point replies to reviewer's comments

Since the comments from reviewer are not listed point by point, here we summarize them into five main points and address them one by one.

1. The contradiction between ATPase activity of ancient KaiC and circadian phenotype:

Re: In *Synechococcus elongatus* PCC 7942, the key circadian clock protein KaiC has multiple catalytic activities, at least including autokinase, autophosphatase and ATPase, which are tightly coupled to fulfill its function in circadian clock. Kondo and co-workers have proposed the hypothesis that the ATPase activity of KaiC is the core rate-limiting reaction for its circadian period length, which is very slow and is temperature compensated [Axmann et al. J Bacteriol 2009;191:5342-5347; Terauchi K, et al. Proc Natl Acad Sci U S A. 2007;104(41):16377-16381; Abe J, et al. Science. 2015;349(6245):312-316].

There are a group of abundant cyanobacterial species contain no KaiA gene, which have no self-sustained rhythms [Holtzendorff et al. J Biol Rhythms 2008;23:187-199; Ma et al. PLoS Genet. 2016; 12:e1005922; Min et al. FEBS Lett. 2005;579:808-812]. For instance, the KaiC protein of marine *Prochlorococcus* sp. strain MED4 which lacks of KaiA, shows a comparable ATPase activity to that of *Synechococcus elongatus* PCC 7942, but *Prochlorococcus* sp. strain MED4 is arrhythmic [Axmann IM, et al. J Bacteriol. 2009;191:5342-5347]. Therefore, the correlation between ATPase activity and circadian period may only apply to the free-running clock systems instead of the hourglass-like timer systems.

It is sort of likewise in *Neurospora crassa*, which is also a model for circadian clock research. Usually, it is acknowledged that the phosphorylation, turnover of FRQ protein and circadian period length are tightly correlated. The higher phosphorylation of FRQ often leads to faster FRQ degradation and shorter period as consequence while lower phosphorylation of FRQ often leads to slower FRQ degradation and longer period [Brown et al. Dev Cell. 2012; 22:477-487; Liu Y, et al. Proc Natl Acad Sci USA. 2000; 97:234-239; Baker CL, et al. Mol Cell. 2009; 34:354-363; Tang CT, et al. Proc Natl Acad Sci USA. 2009; 106:10722-10727], although there are some exceptions [Larrondo et al. Science. 2015;347:1257277; Liu X, et al. Nat Commun. 2019;10:4352]. However, this “rule” may be only applicable in those strains with circadian rhythms instead of those arrhythmic strains. For instance, the strains FRQ Δ (71-77) and FRQ Δ (78-99) bearing deletions of 71-77 and 78-99 regions, showed decreased and increased FRQ stability, respectively. However, both FRQ Δ (71-77) and FRQ Δ (78-99) are arrhythmic [Chen et al. J Biol Chem. 2023;299:104597]. These findings also suggest that the correlations between the certain properties of clock components and circadian period lengths only work in rhythmic systems.

In the ancient KaiABC system, not only the enzymatic activities of ancient KaiC differ from those of KaiC, but also ancient KaiA and KaiB contain differences in sequences and functions from their contemporary counterparts, which means that it is possible that the ancient KaiABC system is not self-sustained although the ancient KaiC display the activities as autokinase, autophosphatase and ATPase.

2. About the “adaptation” of ancient hourglass-like clock system

Let’s suppose the depletion time of an hourglass from the upper part of an hour sand is one hour. If it is turned upside down every 40 minutes, or 80 minutes, anyway, deviated from one hour, since after the ending of every cycle, the sand in the upper part cannot be depleted, it will be impossible to use it to calculate time anymore (Figure 1 below). Therefore, the hourglass-like clock systems may act as timers for certain time lengths when it resonates with the environmental cycles, for organisms to get a time check once a day, despite the lack of free-running property [Mullineaux and Stanewsky R. J Bacteriol. 2009;191(17):5333-5335].

Figure 1. Diagram showing different hour sand-like rhythms aligned to (top panel) and misaligned to the LD cycles (bottom panel), respectively. The sand denoting certain clock components cannot be depleted after the running of each cycle when it is misaligned.

In the cyanobacteria *Prochlorococcus* sp. strain MED4 with only *KaiBC* genes, living in cycling period, KaiC shows diurnal rhythms in its protein levels [Chew J, et al. Nat Commun. 2018;9:3004]. In addition, in the free-living environment, nearly half transcripts of *Prochlorococcus* display diurnal patterns [Ottesen EA, et al. Science. 2014;345(6193):207-212]. The hourglass-type clocks are unidirectional processes [Rensing L, et al. Chronobiol Int. 2001;18(3):329-369]. Thus, the hourglass-type clocks can function as timers although they contain no entire clock machinery and cannot produce self-sustained oscillations, otherwise, the metabolic processes will be chaotic.

How hourglass-like clocks benefit the adaptation certainly is a very important issue for understanding the evolution of circadian clock (including the “throwback” in some bacteria

like *Prochlorococcus*) and environmental adaptation. However, it is a different main topic from the present work although they are related. And to our surprise, despite its importance, this issue has been barely investigated and the underlying molecular mechanisms remain largely unknown. One or mutual interconnections of the dynamic alterations of KaiC ATPase activity, phosphorylation, and turnover rate may be involved in determining the time length which benefits the adaptation to environmental cycling period.

We also feel interested in the investigation of this issue in the future, and during revision we have also conducted two more experiments which provided more clues for understanding of this issue (described in #3).

3. About further required experiments:

Re: According to the comments, we have designed and conducted two additional experiments:

(1) Comparison of expression levels and phosphorylation patterns of KaiC/anKaiC under LD12:12 and LD9:9 on days 13 and 14 after growth in liquid culture

We agree with the reviewer that the strain with the ancestral genes should show fitness vs. environmental period that sharply peaks around an 18 h day. Since we have observed the higher growth rate after 13-d growth (Fig. 6A-F & Fig. EV4 in the manuscript), we collected the samples of strains harboring ancient anKaiABC (*ankaiABC*) and contemporary KaiABC (*kaiABC*), respectively, every 4 h continuously in day 13 and day 14 after growth in liquid culture under LD12:12 and LD9:9. The results showed that the *anKaiABC* strain showed no overt rhythmicity under either LD12:12 or LD9:9; in contrast, the *kaiABC* strain was rhythmic under LD12:12 while the rhythmicity was lost under LD9:9 (Fig. 6G,H in the manuscript). These data demonstrate that the *kaiABC* strain does not fit to the LD9:9 condition.

(2) Comparison of the degradation rates of KaiC and anKaiC

In vivo, not only the ATPase activity and phosphorylation, but also the assembly rate [Emberly E and Wingreen NS. Phys Rev Lett. 2006;96(3):038303], turnover rate, and DNA binding capacity [Mullineaux and Stanewsky R. J Bacteriol. 2009;191(17):5333-5335] could dictate circadian period which correlates with the adaptation to daily cycling environmental cues. Although the ATPase activity of ancient KaiC is much lower than contemporary KaiC, other properties of ancient KaiC may contribute to the adaptation to short day length.

We used chloramphenicol to treat the strains of *kaiABC* and *ankaiABC* to inhibit translation and then compared the degradation rates between KaiC and anKaiC under constant light (LL), to measure and compare the turnover rates of ancient KaiC and KaiC, since the growth experiments showed a better growth and higher levels of photosynthetic pigments

phycobilin and chlorophyll a (Fig. 6A-F in the manuscript). The results showed that anKaiC is degraded at a faster rate (Fig. 6I in the manuscript). Together, these data suggest that although the correlation between ATPase activity and circadian period may not apply to the primordial anKaiABC system which acts in an hourglass-like fashion, the higher turnover rate of anKaiC may contribute to its adaptation under short day length (Fig. 6I in the manuscript).

4. Idiosyncratic to the particular ancestral sequence selected from maximum likelihood estimation or whether these conclusions are robust when considering other nodes on the tree.

Re: In terms of the sequence chosen from the nodes in phylogenetic trees, we selected the sequences from the nodes of 0.95 billion years ago because ancient KaiABC proteins have nodes at the same time (Figure 1A-C in the manuscript). Although they also have common nodes around 0.5 billion years ago or even later, we wanted to investigate the more ancient clock system of cyanobacteria. In addition, if the time of the nodes is closer to today it may be hard to distinguish the difference between the ancient and contemporary clock systems since the daily light-dark cycling period is much closer to that of today.

To improve the expression, we would like to modify the expression to emphasize this point as "...and the results indicated that the ancestral KaiABC sequences have ancestors at the same time ~ 0.95 Ga ago on the basis of the available KaiABC sequences from modern cyanobacterial species...".

5. About the title:

Re: The potential mechanism underlying the adaptation of ancient *kaiABC* strain to LD9:9 condition is really intriguing. Although currently we do not know the mechanism in depth, we have conducted two more experiments which provided some more clues for understanding the potential mechanisms underlying the adaptation of ancient *kaiABC* strain to LD9:9 condition (described in #3), it may be fine to keep using the present title.

Dear Dr. Guo,

Thank you for submitting your revised manuscript (EMBOJ-2024-117992R1) to The EMBO Journal, and for your patience during re-review. Your manuscript has now been seen again by referee #2, and we have received their comments, which you can find below.

As you will see, the reviewer recognizes that the manuscript makes a valuable contribution to the field. However, he/she also points out that overinterpretation of the results and any claims/statements that are not fully supported by the data must be toned down as appropriate. We have discussed this criticism in our editorial team and agree with the referee. I would thus like to kindly ask you to carefully rephrase the title and all relevant claims throughout the manuscript, in line with the referee's suggestions, to make sure that no misleading statements remain in the manuscript. If there are any claims not fully supported by the presented data, please make sure that they are phrased in a way that will accurately reflect their speculative/hypothetical nature. When you are ready to resubmit the final revised version of your manuscript, please include a response to the referee also describing any changes to the final version of the manuscript.

I would also like to ask you to include in your Data availability statement the permanent links (URLs) to your deposited data. Please make sure that all datasets will be publicly available at the time of publication.

We look forward to seeing the final version of your manuscript as soon as possible. Please let us know if you have any questions and use this link to submit your revision: <https://emboj.msubmit.net/cgi-bin/main.plex>.

Best regards,

Ioannis

Referee #2:

I agree with the authors that the question of how ancient KaiC systems functioned and how extant KaiC systems function is interesting and important. Much is not known, and there are valuable data in this manuscript.

However, I feel that the central claims and title of the manuscript are presented in a way that will be misleading to non-specialists. Most readers, seeing the title "Ancient cyanobacterial proto-circadian clock adapted to short daily period ~ 0.95 billion years ago" will assume that the reconstructed ancestral proteins 1.) show phosphorylation cycling in short day-night cycles analogous to the modern circadian clock and 2.) confer a fitness benefit in short day-night cycles relative to genes being absent (*kaiABC*-null).

But in fact neither of these statements are supported by evidence. Instead the ancestral reconstruction seems to have produced a non-oscillating (even in LD cycles) system that would be outcompeted in lab conditions by a strain without the gene cluster. I encourage the authors to avoid overinterpretation.

Point-by-point replies to reviewer's comments

I agree with the authors that the question of how ancient KaiC systems functioned and how extant KaiC systems function is interesting and important. Much is not known, and there are valuable data in this manuscript.

However, I feel that the central claims and title of the manuscript are presented in a way that will be misleading to non-specialists. Most readers, seeing the title "Ancient cyanobacterial proto-circadian clock adapted to short daily period ~ 0.95 billion years ago" will assume that the reconstructed ancestral proteins 1.) show phosphorylation cycling in short day-night cycles analogous to the modern circadian clock and 2.) confer a fitness benefit in short day-night cycles relative to genes being absent (*kaiABC*-null).

But in fact neither of these statements are supported by evidence. Instead the ancestral reconstruction seems to have produced a non-oscillating (even in LD cycles) system that would be outcompeted in lab conditions by a strain without the gene cluster. I encourage the authors to avoid overinterpretation?

Re: We agree with the reviewer's comments, and we have revised the title and related contents in the text accordingly, which include:

- 1) The title: the title has been revised to be "Reconstruction of an ancient cyanobacterial proto-circadian clock occurring ~ 0.95 billion years ago" (pp1, lines 1-2), compared to the previous one ("Ancient cyanobacterial proto-circadian clock adapted to short daily period ~ 0.95 billion years ago"), the updated title is more objective and appropriate.
- 2) The last sentence in abstract has been changed to be "These findings suggest that the ancient cyanobacterial proto-circadian system may not have been endogenously maintained but could have rendered capability for adaptation under the 18-h light/dark cycles ~ 0.95 billion years ago" (pp2, lines 36-38).

Since the last sentence in abstract has been changed, the total number of characters became more than the limitation. Therefore, we also reworded the second sentence in abstract to be "We analysed the ancestor KaiABC (*anKaiABC*) genes and reconstructed the ancient cyanobacterial circadian clock genes that existed circa 0.95 billion years (Ga) ago when the daily light-dark cycle was ~ 18 h" (pp2, lines 26-29).

- 3) The last paragraph in the discussion has been reworded as "Together, these data suggest that the ancient cyanobacterial circadian components already had certain functions in regulating growth or metabolism despite the lack of endogenous oscillation. The ancient proto-circadian clock system may be beneficial for cyanobacterial survival in the 18-h daily cycle ~ 0.95 Ga ago" (pp13, lines 355-358).

Point-by-point replies to Editor's comments

1. I would also like to ask you to include in your Data availability statement the permanent links (URLs) to your deposited data. Please make sure that all datasets will be publicly available at the time of publication.

Re: We have added the link in Data availability statement (pp21, lines 572-573), and all the datasets will be publicly available.

Dear Dr. Guo,

Congratulations on an excellent work! I am very pleased to inform you that your manuscript has now been accepted for publication in The EMBO Journal. Thank you for your thorough responses to the referees' concerns and for addressing our editorial and formatting requests.

If you have any questions, please do not hesitate to contact the Editorial Office. Thank you for your contribution to The EMBO Journal. Working with you has been a pleasure!

Best regards,

Ioannis
